# The translocation assembly module (TAM) catalyzes the assembly of bacterial outer membrane proteins in vitro

Xu Wang [1], Sarah B. Nyenhuis [2] & Harris D. Bernstein [1] ✉

The translocation and assembly module (TAM) has been proposed to play a crucial role in the assembly of a small subset of outer membrane proteins (OMPs) in Proteobacteria based on experiments conducted in vivo using *tamA* and *tamB* mutant strains and in vitro using biophysical methods. TAM consists of an OMP (TamA) and a periplasmic protein that is anchored to the inner membrane by a single α helix (TamB). Here we examine the function of the purified *E. coli* complex in vitro after reconstituting it into proteoliposomes. We find that TAM catalyzes the assembly of four model OMPs nearly as well as the β-barrel assembly machine (BAM), a universal heterooligomer that contains a TamA homolog (BamA) and that catalyzes the assembly of almost all *E. coli* OMPs. Consistent with previous results, both TamA and TamB are required for significant TAM activity. Our study provides direct evidence that TAM can function as an independent OMP insertase and describes a new method to gain insights into TAM function.

Bacteria have evolved a complex cell envelope that maintains cell shape and stability, provides protection from environmental stress, and facilitates nutrient uptake[1]. The cell envelope of Gram-negative bacteria consists of an inner membrane (IM), an outer membrane (OM), and an enclosed aqueous compartment known as the periplasm. The OM is a unique asymmetrical lipid bilayer in which the inner leaflet contains phospholipids and the outer leaflet contains a glycolipid called lipopolysaccharide (LPS)[2]. Outer membrane proteins (OMPs) are also unusual in that they integrate into the lipid bilayer via a 'β barrel', a closed cylindrical structure composed of an even number (8–36) of amphipathic β strands arranged in an antiparallel fashion[3–5]. Despite their common architecture, OMPs are highly diverse. While some OMPs consist solely of an empty β barrel, other OMPs also have a polypeptide embedded inside the barrel lumen and/or extracellular or periplasmic domains[3]. A subset of OMPs also form homodimers or homotrimers[3].

OMPs are synthesized in the cytoplasm and transported across the IM through the Sec machinery[6]. During or shortly after their translocation, various periplasmic chaperones including SurA, Skp,

OsmY, and DegP bind to the newly synthesized OMPs to escort them through the periplasm while maintaining them in an insertion-competent state and potentially promoting partial folding before their membrane integration[7–15]. These chaperones have partially redundant but also distinct functions. SurA is thought to be the most important chaperone because the integrity of the OM is impaired and the steady-state level of OMPs is significantly reduced in *surA-* strains[8,9]. Skp forms a jellyfish-like homotrimer with a central cavity that can accommodate unfolded or partially folded OMPs to prevent aggregation and to target OMPs with assembly (folding and membrane insertion) defects for degradation[16–19]. OsmY is involved in the biogenesis of an OMP family known as diffuse adherence autotransporters and protects them from proteolysis[12]. DegP acts primarily as a protease and exhibits chaperone activity at low temperatures[20,21].

Once most OMPs reach the OM, their assembly is catalyzed by a nanomachine known as the β-barrel assembly machine (BAM) that is universal in Gram-negative bacteria[22]. In *E. coli*, BAM consists of five proteins, a 16-stranded β-barrel protein (BamA) that is linked to five periplasmic polypeptide transport-associated (POTRA) domains, and

[1]Genetics and Biochemistry Branch, National Institute of Diabetes and Digestive and Kidney Diseases, National Institutes of Health, Bethesda, MD 20892, USA. [2]Laboratory of Cell and Molecular Biology, National Institute of Diabetes and Digestive and Kidney Diseases, National Institutes of Health, Bethesda, MD 20892, USA. ✉e-mail: harris_bernstein@nih.gov

four lipoproteins (BamB-E) that are bound to the POTRA domains[22–26]. Only BamA and BamD are conserved and essential, although the presence of all subunits is optimal for BAM function[27,28]. Unlike the vast majority of β barrels that are held together by substantial hydrogen bonding between the first and last β strands and are therefore highly stable, the BamA β barrel seam is held together by only a few hydrogen bonds and is consequently inherently unstable[24–26]. It was proposed that the transient opening of the BamA barrel plays a key role in OMP membrane integration partially by destabilizing the local lipid bilayer[29–31]. Disulfide bond crosslinking experiments showed that the open form of the BamA β barrel forms a hybrid barrel with incoming OMPs through (1) a tight interaction between the first BamA β strand [BamA(β1)] and the conserved "β signal" motif (GXXϕXϕ, where ϕ is an aromatic amino acid) located in the C-terminal β strand of the client and (2) weaker, dynamic interactions between the C-terminal BamA β strands [BamA(β15-β16)] and the N-terminal β strand of the client[32,33]. Recent cryo-EM studies have corroborated these results and shown that after interacting with BamA, the client progresses from a curved β sheet conformation to a barrel structure before it ultimately dissociates through a strand-exchange mechanism[34,35]. The discovery of a natural product (darobactin) that has potent bactericidal activity by functioning as a competitive inhibitor of β signal binding further underscores the importance of the BamA(β1)-β signal interaction in the OMP assembly process[36,37].

Interestingly, a second nanomachine that also appears to play an important role in the assembly of at least a few *E. coli* OMPs is called the translocation and assembly module (TAM)[22,38]. Unlike BAM, TAM consists of only two subunits, a 16-stranded β-barrel OMP (TamA) and a ~137 kDa protein (TamB) that is anchored to the IM by a predicted N-terminal α helix[38,39]. TamA is evolutionarily related to BamA (both proteins are members of the Omp85 superfamily) but is found almost exclusively in Proteobacteria and Bacteroidetes[38–41]. Although the sequences of *E. coli* K-12 BamA and TamA are only 21% identical, their β barrels are structurally closely related[24–26,42,43]. TamA has a putative lateral gate between its first and last β strands like BamA and three POTRA domains[24,42–45]. Several studies have provided experimental evidence that residues that are located near the C-terminus of TamB bind to the N-terminal POTRA domain of TamA[38,44,46,47]. A recent in silico analysis of TAM structure performed with AF2Complex, however, predicts that the C-terminal ~70 residues of TamB fold into a partial β-barrel that mimics an OMP by binding to the first β strand of a laterally open form of TamA[48]. TamB residues that are just upstream of this segment are also predicted to be in close proximity to the TamA POTRA domains. TAM is 20–40 times less abundant than BAM in *E. coli*[49,50], and the deletion of *tamA* and/or *tamB* produces almost no discernable phenotype under laboratory growth conditions[39,51]. Nevertheless, TAM is necessary for the efficient biogenesis of several OMPs, including the *Citrobacter* autotransporter p1121 (when it is expressed in *E. coli*), two adhesins Antigen 43 (Ag43) and EhaA, usher proteins such as FimD and UshC, an efflux pump TolC, and the intimins EaeA and FdeC[38,43,44,46,47,52–54]. TAM is also required for virulence or colonization in a variety of organisms[38,45,55–63]. Although the mechanism by which TAM catalyzes OMP assembly is unclear, several models have been proposed in which TAM functions independently of BAM, collaborates with BAM, or acts at a different stage of OMP biogenesis[64]. Indeed, the observation that locking the TamA lateral gate blocks the TAM-mediated biogenesis of FimD suggests that the opening of the barrel is critical for function[43]. Curiously, unlike TamA, TamB is widely distributed in Gram-negative bacteria[40]. X-ray crystallography of a portion of a highly conserved C-terminal domain of unknown function (DUF490) revealed a 'β-taco' shaped segment containing a hydrophobic groove that is predicted to span the entire length of TamB[65]. The finding that TamB interacts with BamA in *Borrelia burgdorferi* and that *tamB*-like genes are often found immediately downstream of *bamA* in the same operon is consistent with the idea that TamB plays a

role in OMP assembly in at least in some organisms[40,66]. It is very noteworthy, however, that recent evidence strongly suggests that TamB, which is a member of the AsmA-like family of proteins[40], works together with TamA to promote phospholipid transport to and from the *E. coli* OM[67,68].

To test the ability of TAM to catalyze OMP assembly independently from BAM, we developed a novel method to purify and reconstitute TAM into lipid vesicles in vitro. We found that in the presence of the TAM proteoliposomes, several different urea denatured OMPs were assembled about as efficiently as they were in the presence of BAM proteoliposomes generated by the same method. We also found that proteoliposomes containing TamA alone were much less active than proteoliposomes that contained both TamA and TamB. Finally, we obtained evidence that disrupting the interaction between TamA(β1) and the last β strand of the client protein significantly reduced assembly efficiency, suggesting that BAM and TAM function by similar mechanisms. Although our results do not rule out the possibility that TAM has multiple functions, they are consistent with a model in which TAM is sufficient to catalyze OMP assembly. Furthermore, our novel TAM reconstitution and in vitro folding assay provide a method to identify putative TAM cofactors and potential substrates and to enable us to gain additional insights into the broader functions of TAM and its mechanism(s) of action.

## Results
### Purification of TAM and its reconstitution into proteoliposomes
A remarkable degree of overlap in the *E. coli* BamA and TamA β barrels was previously observed in a study in which published crystal structures were superimposed (Fig. 1a)[47]. Although most of the loops in the TamA β barrel are shorter than their cognate loops in the BamA β barrel, the calculated RMSD of their β-strand backbones is 2.078 Å (Supplementary Table 1). The fundamental similarity in the overall structure of the TamA and BamA β barrels in itself suggests that the two proteins have similar functions.

Based in part on this observation, we sought to develop an in vitro assay to study the function of TAM using purified components. Our goal was to optimize the purification of active TAM and to reconstitute the complex into proteoliposomes (Fig. 1b). We found that the method used to purify TAM greatly affected its purity and functionality. We initially generated plasmid pXW47 [$P_{trc}$-(8xHis)-TamAB] to express TAM with an N-terminal His-tag attached to TamA for purification by affinity chromatography. Based on previous studies[38,44], we assumed that TamB would co-purify with TamA. Because we found that TamB was somewhat susceptible to proteolysis and that the amount of co-purified full-length TamB significantly affected TAM activity, a second copy of *tamB* with an independent ribosome binding site was added to generate expression plasmid pXW48 [$P_{trc}$-(8xHis)-TamAB-TamB].

We produced TAM in *E. coli* BL21-CodonPlus (DE3)-RIPL, a strain that harbors the rare codon tRNAs needed for maximum *tamB* expression. After cells were transformed with pXW48, TAM synthesis was induced by adding IPTG (isopropyl β-D-1-thiogalactopyranoside) to cultures and incubating them at 16 °C overnight. Cells were lysed by a cell disrupter. Other cell lysis methods, including sonication and the addition of protein extraction reagents, were not optimal because they led to the loss of TamB or the co-purification of a considerable amount of non-specific contaminants. We found that protease inhibitors, such as cOmplet protease inhibitor cocktail (Roche), phenylmethylsulfonyl fluoride (PMSF), and β-mercaptoethanol, significantly decreased TAM activity and thus could not be used. After cell lysis, membranes were pelleted by ultracentrifugation. To solubilize membrane proteins, we found that 1% n-dodecyl-beta-maltoside (DDM) in a PBS buffer was optimal. Curiously, glycerol and a detergent lauryldimethylamine N-oxide (LDAO) also disrupted TAM activity.

After binding His-tagged TAM to Ni-NTA agarose and analyzing eluted proteins by SDS-PAGE and Coomassie blue staining, we

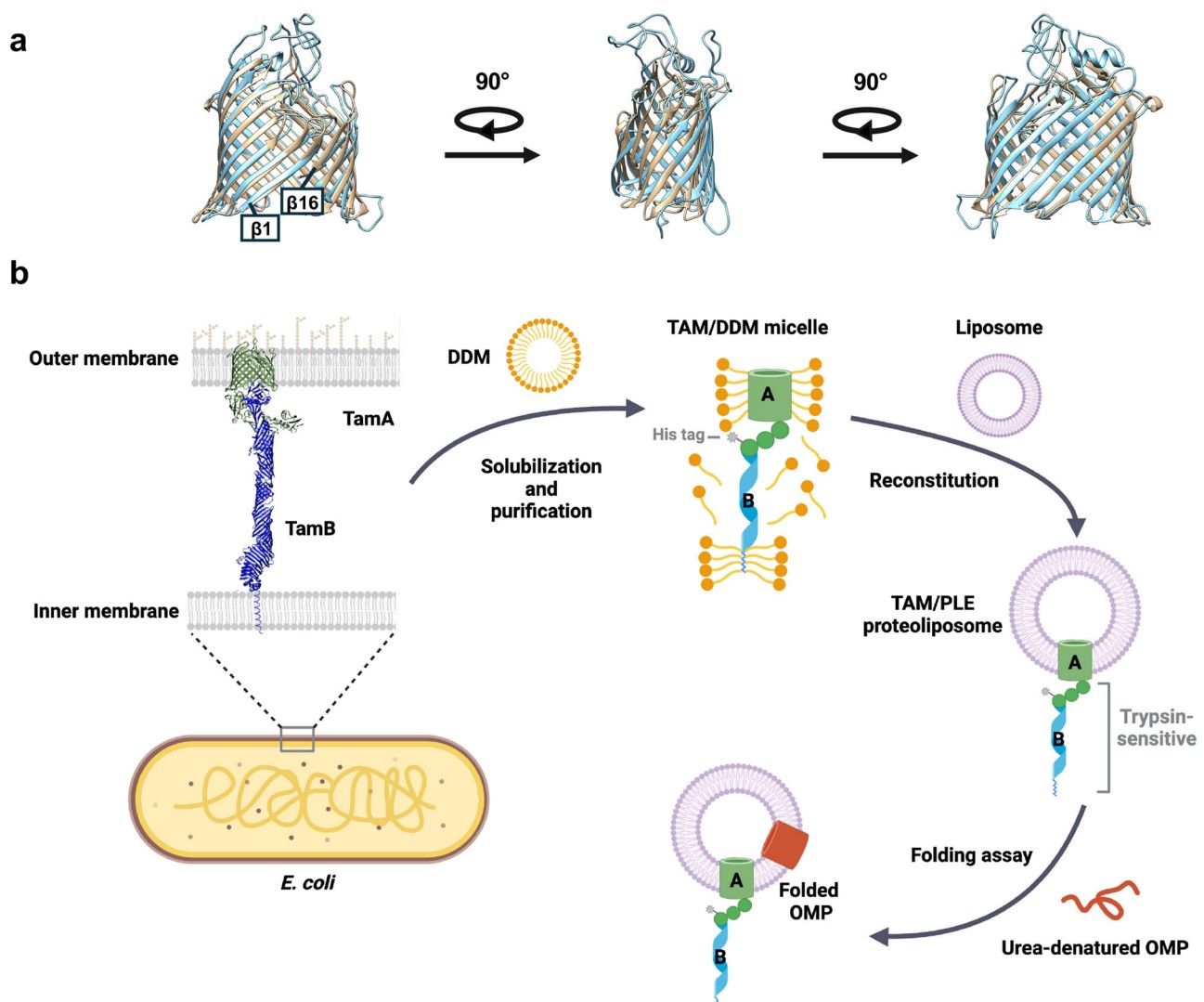

**Fig. 1 | Structural comparison of TamA and BamA β-barrels and outline of OMP assembly assay. a** Structural superimposition of TamA (PDB: 4C00[42], residues 266–573, beige) and BamA (PDB: 8BVQ[92], residues 424–807, light blue) β-barrels. The calculated RMSDs are listed in Supplementary Table 1. **b** Outline of the TAM-mediated OMP assembly assay used in this study. The *E. coli* TAM [His8-TamA (PDB: 4C00] and TamB (based on the structure predicted by AlphaFold2, AF2Complex and MD simulations[48,109–111]) was expressed in vivo, solubilized from a total membrane fraction with n-dodecyl-β-D-maltoside (DDM), and purified on Ni-NTA agarose. The purified TAM was reconstituted into *E. coli* polar lipid extract (PLE)

liposomes, and the predicted topology was validated by trypsin digestion. Based on the results of the trypsin digestion, it is unclear if the N-terminal α-helix of TamB is integrated into a separate liposome (see Source Data file, p. 1). In OMP folding assays, urea denatured OMPs were incubated with the proteoliposomes containing TAM, and their folding and integration into the vesicles was assessed by determining the percent of the protein that was resistant to SDS denaturation in the absence of heat. Part **b** was created with BioRender.com, released under a Creative Commons Attribution-NonCommercial-NoDerivs 4.0 International license.

observed bands corresponding to TamB (~137 kDa) and TamA (~63 kDa) (Fig. 2a), suggesting that TamA and TamB were co-purified as expected. The DDM-solubilized TAM was reconstituted into pre-formed liposomes derived from an *E. coli* polar lipid extract (PLE) by rapid dilution (Fig. 1b), and the vesicles (hereafter referred to as TAM/PLE proteoliposomes) were pelleted by ultracentrifugation. Most of the TAM was found in the pellet but remained in the supernatant in the absence of liposomes (Fig. 2b), and therefore was likely reconstituted into the vesicles. Consistent with previous results[38,44], the unheated TAM/PLE proteoliposomes migrated slightly slower than 480 kDa on blue native PAGE, whereas after heating, a ~150 kDa band corresponding to TamA was observed (Supplementary Fig. 1). To confirm that TAM was properly reconstituted, the proteoliposomes were treated with trypsin (Fig. 1b). As expected, a band corresponding to the TamA β-barrel (~35 kDa) was observed after tryptic digestion (Fig. 2c, lane 2), suggesting that the TamA POTRA domains and TamB were

exposed on the surface and degraded by trypsin while the TamA β-barrel was inserted into the lipid bilayer and thereby protected from the protease (Fig. 1b).

Using the protocols described above that were devised to purify TAM and to produce TAM/PLE proteoliposomes, we also produced proteoliposomes containing either His-tagged TamA or BAM with a His-tag on BamE in the same inside-out orientation (Supplementary Fig. 2). We were unable, however, to purify His-tagged full-length TamB or TamB fragments under conditions that do not interfere with TAM function. Furthermore, when we introduced a pair of cysteine residues into TamA β1 and β16 that form a disulfide bond that locks the lateral gate, the level of TamB that was co-purified with TamA was reduced dramatically and matched the small fraction of TamA that did not undergo disulfide bond formation (Supplementary Fig. 3).This observation is consistent with the prediction that the C-terminus of TamB interacts with TamA β1[48]. To confirm the purity of TAM and BAM, we

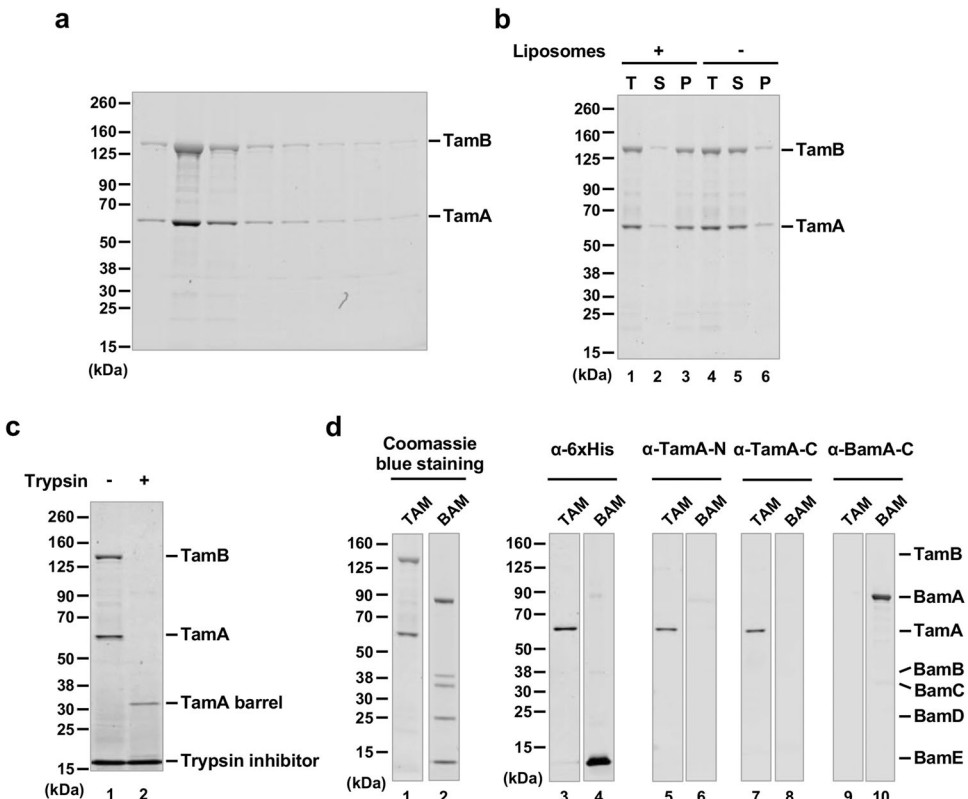

**Fig. 2 | Purification and reconstitution of TAM. a** SDS-PAGE analysis of fractions eluted from Ni-NTA agarose. **b** After purified TAM was mixed with PLE liposomes, the total mixture (T) was ultracentrifuged to separate the supernatant (S), which contains free TAM, and the pellet (P), which contains reconstituted TAM (lanes 1–3). The fate of TAM in the absence of liposomes was examined in parallel (lanes 4–6). Samples were analyzed by SDS-PAGE and Coomassie blue staining. **c** TAM/PLE proteoliposomes were treated with trypsin to digest exposed proteins and protein segments or untreated. The reaction was stopped by adding a trypsin inhibitor (Millipore Sigma, catalog number 10109886001) and analyzed by SDS-PAGE. Proteins were visualized by Coomassie blue staining. TAM was purified and reconstituted into proteoliposomes ten times with results that are similar to those shown in **a**–**c**. **d** TAM or BAM proteoliposomes (see Supplementary Fig. 2) were analyzed by SDS-PAGE and Coomassie blue staining or Western blot using the indicated antiserum. This comparative analysis was conducted twice with similar results.

analyzed TAM/PLE and BAM/PLE proteoliposomes by Coomassie blue staining (Fig. 2d, lanes 1–2) and Western blotting using multiple antisera. Using an anti-His-tag antibody, we detected TamA and BamE (~12 kDa) only in TAM/PLE and BAM/PLE samples, respectively (Fig. 2d, lanes 3–4). Likewise, using antisera raised against TamA N- and C-terminal peptides we detected TamA only in TAM/PLE (Fig. 2d, lanes 5–8), and using an antiserum raised against a BamA C-terminal peptide we detected BamA only in BAM/PLE (Fig. 2d, lanes 9–10). These results strongly suggest the TAM activity discussed below can be attributed to TAM itself and not to any BAM contaminants.

## TAM catalyzes the assembly of OMPs in vitro
Using the TAM/PLE proteoliposomes, we wished to gain insight into TAM function. To determine if TAM can catalyze OMP assembly, we set up in vitro experiments to study the folding of four OMPs (Supplementary Fig. 4a). One model protein, OmpA, has an 8-stranded empty β barrel domain and a periplasmic domain[69]. The second protein, EspPΔ5′ [previously designated EspP (β + 46)[70]], is a truncated form of the autotransporter EspP that retains the 12-stranded β barrel and a His-tagged linker that traverses the β barrel but lacks the extracellular passenger domain[70–73]. EspPΔ5′ has been shown to fold as efficiently as full-length EspP in vivo and in vitro[18,70,73]. The third protein is a naturally cleaved C-terminal fragment of the autotransporter Antigen 43 (Ag43-β, residues 552-1039) that contains a 12-stranded β barrel and a portion of the extracellular autochaperone (AC) domain[12,74,75]. The last protein is the long-chain fatty acid porin FadL, which folds into a 14-stranded β barrel[76]. Each protein was produced in *E. coli*, purified in inclusion bodies, and solubilized in a buffer containing 8 M urea.

In our OMP folding assay, one of the urea denatured OMPs was mixed with TAM/PLE, TamA/PLE, or BAM/PLE proteoliposomes or empty PLE liposomes and incubated at 30 °C for 1 h. A roughly equivalent number of proteoliposomes was added by normalizing the molar concentrations of TamA and BamA. Samples were collected before (t = 0.5 min) and after (t = 60 min) the incubation. We monitored protein folding on Western blots by exploiting the 'heat modifiability' of OMPs, a property based on the observation that fully folded OMPs are resistant to SDS denaturation and migrate more rapidly than their predicted molecular weight on SDS-PAGE unless they are heated[12,70,77]. We also assessed the folding of the OMPs by their proteinase K (PK) resistance. The β barrels of completely folded OMPs inserted into a proteoliposome are often resistant to PK digestion, but unfolded proteins and periplasmic domains not protected by the lipid bilayer are often degraded.

Interestingly, we found that TAM (but not TamA alone) could catalyze the assembly of all four OMPs we tested nearly as efficiently as BAM (Fig. 3). Based on Western blots in which we detected OmpA using an antiserum directed against β barrel loop 4 (L4), as much as 90% of the protein folded into the TAM/PLE or BAM/PLE proteoliposomes and migrated rapidly in the absence of heat (Fig. 3a, lanes 3 and 15). The relatively weak signal in the unheated samples (which is often observed when fully folded OMPs are detected by anti-peptide antisera on Western blots[78]) is presumably due to the reduced accessibility of the epitope recognized by anti-OmpA L4. In the heated samples, OmpA was unfolded and migrated at its predicted molecular weight (Fig. 3a, lanes 4 and 16). Curiously, we found that OmpA was primarily resistant to PK treatment even though it contains an accessible periplasmic domain

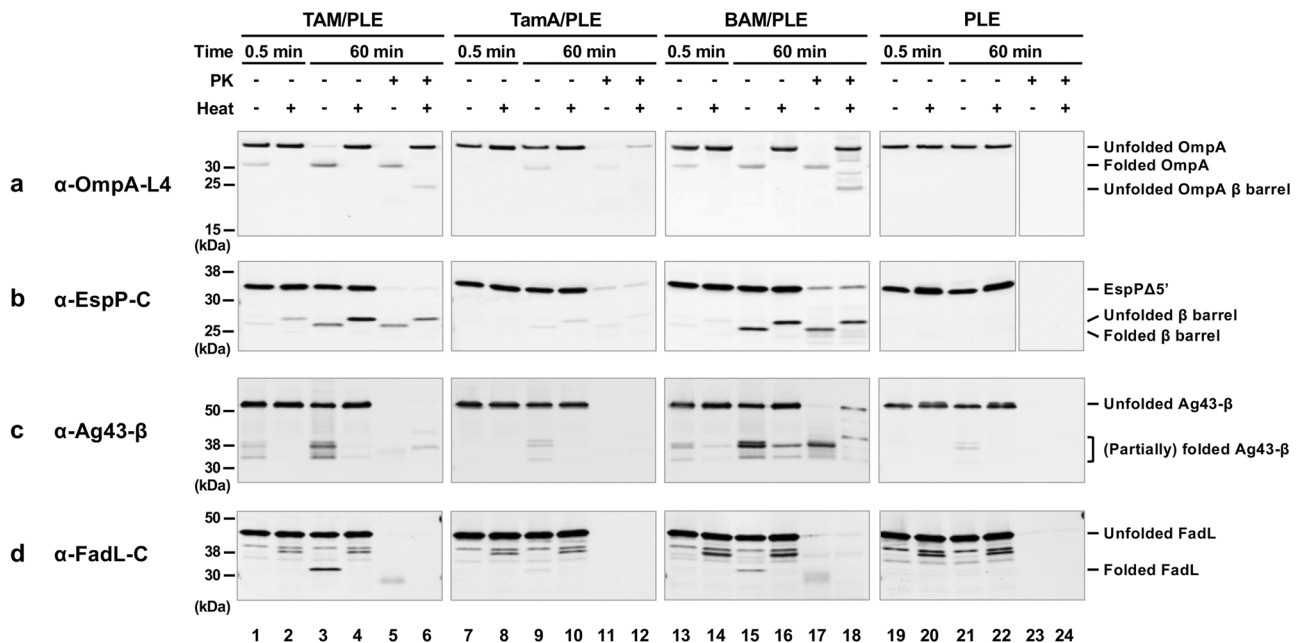

**Fig. 3 | TAM can promote the assembly of OMPs in vitro.** Urea-denatured OmpA (**a**), EspPΔ5' (**b**), Ag43-β (**c**), or FadL (**d**) were incubated with PLE proteoliposomes containing 2 μM TAM, TamA, or BAM or empty PLE liposomes at 30 °C. Samples were collected from each reaction after 0.5 and 60 min, treated with proteinase K (PK), or left untreated. After adding the loading buffer, samples were placed on ice or heated to 95 °C and resolved by SDS-PAGE. OMP folding was assessed by determining the fraction of the protein that was resistant to SDS denaturation in the absence of heat (and that underwent autocatalytic cleavage in the case of EspPΔ5') by Western blot using the indicated antiserum. For clarity, an experiment that shows a more highly resolved view of the folded forms of Ag43-β is depicted in Supplementary Fig. 4b, and an experiment in which the assembly of Ag43-β in the presence of TAM/PLE or TamA/PLE was repeated but the blot was overexposed is shown in Supplementary Fig. 4c. All of the experiments shown here were repeated three times with similar results.

(Fig. 3a, lanes 5–6 and 17–18). As indicated by the presence of a faint ~20 kDa band corresponding to the free β domain, only a fraction of the protein was cleaved by the protease. It should be noted, however, that only a fraction of the OmpA inserted into the OM in vivo is sensitive to PK digestion (Janine H. Peterson and HDB, unpublished data). OmpA was assembled into TamA/PLE proteoliposomes much less efficiently than TAM/PLE proteoliposomes, but the folded protein showed similar heat modifiability and PK-resistance patterns (Fig. 3a, lanes 7–12). The results suggest that TamA is sufficient to catalyze a low level of OMP assembly, but that both TamA and TamB are required to catalyze efficient assembly. To confirm that the assembly of OmpA was catalyzed by TAM and TamA, we measured the amount of BamA and BamD that was present in TAM/PLE and TamA/PLE proteoliposomes and found that the level of BAM contamination (<0.7% of the total protein) was insufficient to catalyze significant OmpA assembly (Supplementary Fig. 5a, b). Furthermore, while PMSF inhibited TAM-mediated assembly of OmpA, consistent with previous results[79,80] the protease inhibitor did not affect BAM activity (Supplementary Fig. 5c). As expected, essentially no folding was seen when OmpA or any of the urea denatured OMPs (see below) were incubated with empty PLE liposomes and all of the protein added to the empty vesicles was PK-sensitive (Fig. 3, lanes 19–24).

In the case of EspPΔ5', we could analyze assembly not only by monitoring heat modifiability, but also by monitoring the autocatalytic release of a 46-residue N-terminal fragment from the β barrel. This proteolytic maturation is a naturally occurring intra-barrel cleavage reaction that occurs only when the β barrel domain is fully assembled (Supplementary Fig. 4a)[81]. In samples that contained TAM/PLE proteoliposomes, approximately 25% of the EspPΔ5' underwent proteolytic maturation as indicated by the detection of a ~25–30 kDa band that corresponds to the free β barrel domain using an antiserum generated against an EspP C-terminal peptide[70] (Fig. 3b, lanes 3–4). Consistent with previous results[70,73], the assembled β barrel migrated more rapidly in the absence of heat and was resistant to PK digestion

(Fig. 3b, lanes 5–6). Similar results were obtained when BAM/PLE proteoliposomes were added to the reaction, although assembly efficiency appeared slightly higher (Fig. 3b, lanes 13–18). In contrast, only very limited assembly was observed when TamA/PLE proteoliposomes were added to the reaction (Fig. 3b, lanes 7–12).

In the presence of TAM/PLE or BAM/PLE proteoliposomes, Ag43-β was folded into at least four forms that migrated between 30 kDa and 40 kDa (Fig. 3c, lanes 3 and 15; Supplementary Fig. 4b). Multiple bands have been previously observed in vivo[12], and it is unclear if these represent alternate conformations of the folded protein (that might result from a tendency of some portions of the unheated protein to unfold in SDS) or self-cleaved forms. After prolonged heating (95 °C, 15 min), a portion of two forms of Ag43-β still migrated rapidly (Fig. 3c, lanes 4 and 16). These two polypeptides might correspond to Ag43-β after the extracellular AC domain was cleaved from the β barrel (the ~35 kDa species) and Ag43-β after both the AC domain and the linker were released (the ~31 kDa species) (Supplementary Fig. 4a)[82]. Curiously, while three fast-migrating Ag43-β species were resistant to PK in the presence of BAM/PLE proteoliposomes, only one species was PK-resistant in the presence of TAM/PLE proteoliposomes (Fig. 3c, lanes 5 and 17). This result might be due to differences in the protection of the protein embedded in the two types of proteoliposomes. Much less assembled Ag43-β was observed in the presence of TamA/PLE proteoliposomes, but the pattern of PK protection was similar to that observed in the presence of TAM/PLE proteoliposomes (Fig. 3c, lanes 7–12; Supplementary Fig. 4c).

Finally, we found that FadL could also be assembled into TAM/PLE and BAM/PLE proteoliposomes based on a rapidly migrating ~32 kDa band that was detected by a C-terminal antiserum in unheated samples (Fig. 3d, lanes 3 and 15). We observed a band of the same size when we synthesized FadL in a coupled in vitro transcription-translation system (the PURE system) that contained BAM/POPC proteoliposomes instead of isolating the protein from inclusion bodies (Supplementary Fig. 3d).

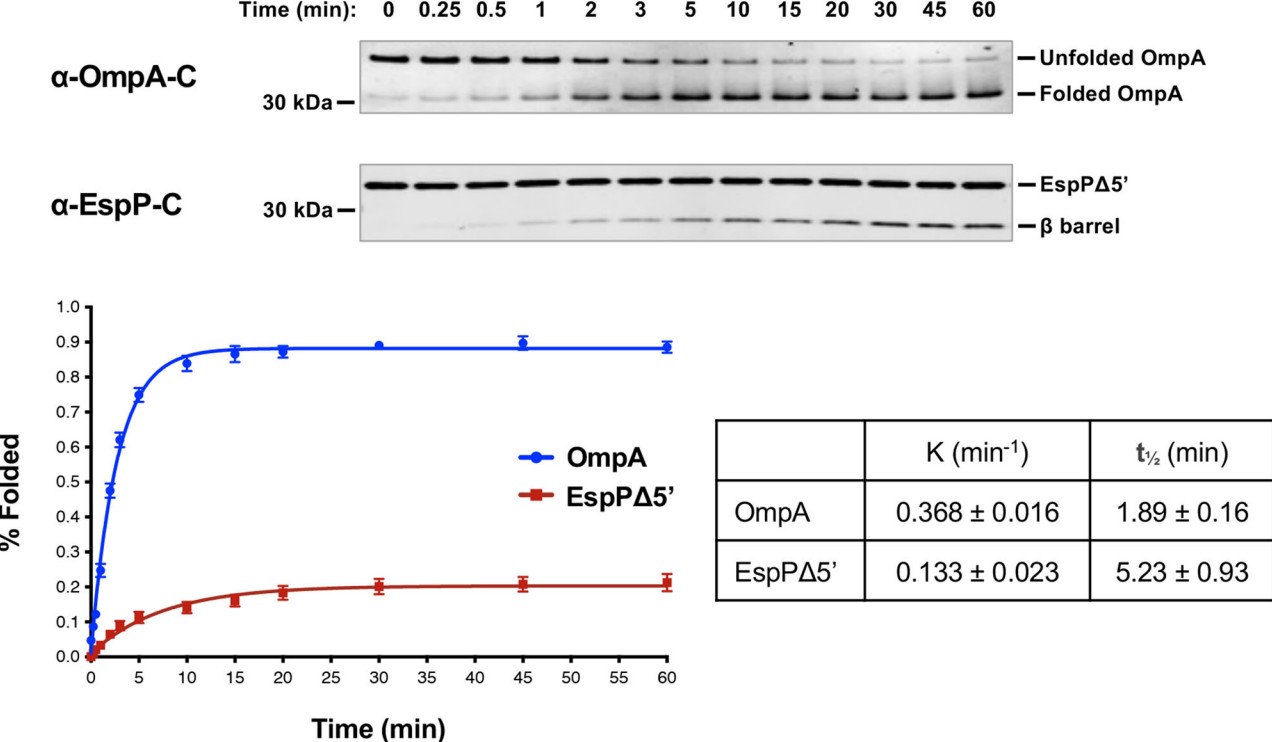

**Fig. 4 | TAM catalyzes rapid OMP assembly in vitro.** Urea-denatured OmpA or EspPΔ5' was incubated with 2 μM TAM/PLE proteoliposomes at 30 °C for up to 60 min. Samples were collected at various time points, mixed with loading buffer, and placed on ice. Unheated proteins were resolved by SDS-PAGE, and OmpA was detected by Western blot using an antiserum raised against a C-terminal peptide. The percentage of folded OmpA was calculated using the formula (folded OmpA/total OmpA) x 100 where folded OmpA was defined as protein that migrated faster than the expected molecular weight. EspPΔ5' samples were mixed with loading buffer, heated to 95 °C, resolved by SDS-PAGE, and detected by Western blot using an antiserum against a C-terminal peptide. Folding was assessed by determining the percent of the protein that underwent proteolytic maturation by using the formula (cleaved β barrel/(cleaved β barrel + uncleaved EspPΔ5') x 100). Representative experiments are shown at the top. The curves on the bottom show the best fit to the data obtained in five independent experiments performed with OmpA and four performed with EspPΔ5'. The data are presented as mean values +/- standard error of the mean. The rate constant (K) and the time required to reach 50% maximal folding ($t_{1/2}$) were calculated using GraphPad Prism 8 software.

Presumably due to a subtle difference in protein structure or proteoliposome architecture, FadL synthesized in the PURE system and assembled into a POPC bilayer was resistant to PK digestion, while the identical protein synthesized in vivo and purified from inclusion bodies was sensitive to PK after assembly into TAM/PLE and BAM/PLE (Supplementary Fig. 4d; Fig. 3d, lanes 5 and 17). Only a very low level of the -32 kDa band was observed in reactions that contained TamA/PLE (Fig. 3d, lanes 7–12).

To gain further insight into TAM function, we next examined the kinetics of assembly. OmpA was not only assembled more efficiently than EspPΔ5', but also more rapidly based on the time required to reach 50% maximal assembly ($t_{1/2}$) (Fig. 4). Indeed EspPΔ5' might be assembled less efficiently than OmpA because the protein loses insertion competence more rapidly than it is inserted into TAM/PLE proteoliposomes. In any case, our results show that, like BAM, TAM catalyzes the assembly of different OMPs at significantly different rates in vitro[73,83].

We used a concentration of proteoliposomes in our experiments (2 μM) that we found to be optimal for both TAM/PLE and BAM/PLE under our experimental conditions. It should be noted, however, that significant TAM-mediated OMP assembly could be observed at the lower proteoliposome concentrations (0.1–0.5 μM) that have been used in previous BAM-mediated in vitro OMP assembly assays[70,73,84] (Supplementary Fig. 6). We also tested different phospholipids for TAM reconstitution, including four synthetic phospholipids (POPC, DPPC, DMPC, and DLPC) that create lipid bilayers of different thickness and fluidity, and found that the TAM/PLE proteoliposomes were not only the most active (Supplementary Fig. 7), but also gave the most consistent results across different protein preps.

## Skp inhibits TAM-mediated OMP assembly in vitro

The results above show that TAM/PLE and BAM/PLE can catalyze OMP assembly without chaperones under our experimental conditions. Nevertheless, we wanted to determine if periplasmic chaperones significantly influence OMP folding and if their effects on assembly differ. In these experiments, we examined the effects of SurA, OsmY, Skp, and DegP. Because DegP has both chaperone and protease activities and degrades OMPs in in vitro folding assays[11,83], we tested DegP$^{S210A}$, a mutant that lacks the protease activity but not the chaperone activity[21]. Urea-denatured OmpA or EspPΔ5', a chaperone, and TAM/PLE, TamA/PLE, or BAM/PLE proteoliposomes were sequentially added to a Tris buffer. The final ratio between the chaperone and proteoliposomes was 1:1. Interestingly, SurA, OsmY, and DegP$^{S210A}$ did not affect the assembly of either OMP by TAM, TamA or BAM (Fig. 5, lanes 1–3, 5, 6–8 and 10; Supplementary Fig. 8), In contrast, Skp significantly decreased the level of folded both OmpA and EspPΔ5' (Fig. 5, lanes 4 and 9; Supplementary Fig. 8). Our data are consistent with the results of previous studies that show that Skp has a unique ability to inhibit the assembly of urea denatured OMPs catalyzed by BAM (presumably by sequestering the proteins or reducing their assembly-competence)[18,83] and are consistent with the possibility that TAM and BAM catalyze OMP assembly by fundamentally similar mechanisms.

## The β signal promotes efficient TAM-mediated OMP assembly

In addition to having structurally similar β-strand backbones and a relatively small number of hydrogen bonds (H-bonds) between β1 and β16 that suggest that *E. coli* TamA and BamA β barrels both have an unstable seam (or "lateral gate"), we noticed striking similarities

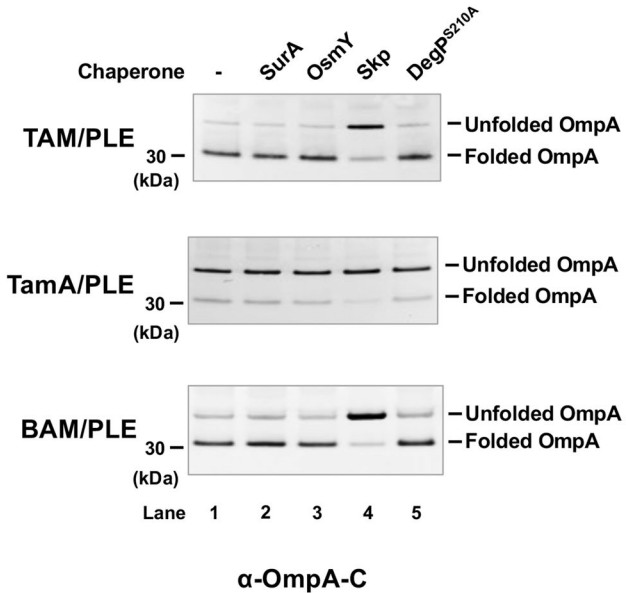

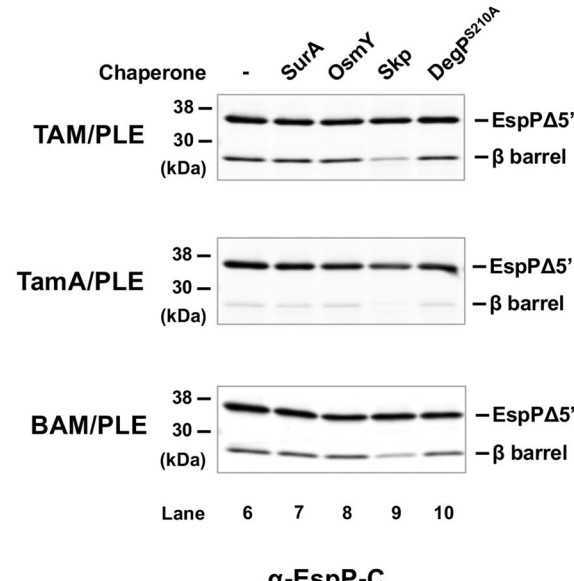

α-OmpA-C

α-EspP-C

**Fig. 5 | Effect of periplasmic chaperones on TAM folding in vitro.** Urea denatured OmpA or EspPΔ5' was incubated with the indicated chaperone (2 μM) and proteoliposomes containing either 2 μM TAM, TamA, or BAM for 60 min at 30 °C. Samples were then unheated (for OmpA experiments) or heated to 95 °C for 10 min (for EspPΔ5' experiments) and resolved by SDS-PAGE. Western blotting was performed using an antiserum against an OmpA or EspP C-terminal peptide. A representative experiment is shown. Three independent experiments were performed and a statistical analysis is shown in Supplementary Fig. 8.

between the sequences of several segments of the two β barrels. The β1 strand of TamA resembles the cognate strand of BamA, which has been reported to form a stable non-sliding interface with the β signal of client proteins during assembly[32] (Fig. 6a, left). Furthermore, in members of the 39 Proteobacterial families in which TamA has been annotated, TamA and BamA share a conserved sequence motif [G(V/A/I)GY(S/G)(Q/T/S)] even though their sequences are typically only 21–46% identical to those of their *E. coli* orthologs (Supplementary Fig. 9a, Supplementary Table 2a). Based on a structural model of a BAM-EspP folding intermediate (PDB: 7NRI[34]), BamA(β1) binds to the EspP β signal [EspP(β12)] through seven backbone H-bonds (Fig. 6B, left panel). Remarkably, despite the fact that the TamA(β1) amino acid sequence is not identical to the BamA(β1) sequence, we used the *E. coli* TamA crystal structure (PDB: 4C00[42]) with Chimera and COOT to predict that TamA(β1) can likewise form seven backbone H-bonds with EspP(β12) without causing any steric clashes (Fig. 6B, right panel). The sequence of *E. coli* TamA(β16) is also very similar to that of *E. coli* BamA(β16). It contains both the key glycine residue that is critical for the formation of a C-terminal kink that modulates the opening of the lateral gate[85] and a tripeptide motif that is similar to the FQF motif that is conserved in BamA homologs (Fig. 6a, middle and Supplementary Fig. 9b)[86]. Finally, the sequence of TamA extracellular loop 6 (eL6), a large loop that penetrates the β barrel lumen and is thought to undergo conformational changes that are associated with the reaction cycle of Omp85 proteins[87,88], is similar to that of BamA (eL6) (Fig. 6a, right). TamA and BamA (both in *E. coli* and other Proteobacteria) not only share the conserved (V/I/L)RG(F/Y) motif that is characteristic of eL6 in multiple Omp85 families[89], but also share many of the surrounding residues (Supplementary Fig. 9c). In contrast, members of the two-partner secretion transporter (TpsB) family, an Omp85 family that catalyzes protein secretion reactions instead of OMP insertion reactions, do not share conserved motifs in either β1 or β16. Furthermore, their eL6 consensus sequence differs more from the cognate consensus sequences of TamA and BamA than the TamA and BamA consensus sequences differ from each other (Supplementary Fig. 9c, Supplementary Table 2b). Taken together, the structural and sequence data suggest that BamA and TamA catalyze OMP assembly by similar mechanisms that involve recognition of the β signal of client proteins.

To examine the hypothesis that TamA binds to the β signal of OMPs during assembly, we mutated the highly conserved aromatic residues at positions −1 and −3 in the β signal motifs of OmpA (GVSYRF) and EspPΔ5 (NFRYSF) to alanine to generate OmpA^Y189A, F191A and EspPΔ5'^YI298A, F1300A. The same mutations were shown to impair BAM-mediated OmpA and EspPΔ5' folding both in vivo and in vitro[18]. OmpA^L98R, V100R and EspPΔ5'^L1119R, which contain surface-exposed arginine mutations that block assembly in vivo and in vitro[18,90], were tested as controls. Based on the observation that β signal mutations can affect the kinetics but not the completion of OMP assembly[18], we incubated wild-type and mutant forms of OmpA or EspPΔ5' with BAM/PLE or TAM/PLE for only a short time (2 min for OmpA; 15 min for EspPΔ5'). Consistent with our hypothesis, the β signal mutations reduced both BAM- and TAM-mediated OMP assembly (Fig. 6c, lanes 1–2 and 4–5; Supplementary Fig. 10a). In samples that contained OmpA^Y189A, F191A, smeared bands that might correspond to partially folded OmpA assembly intermediates were observed (Fig. 6c, lanes 2 and 5; Supplementary Fig. 10a). As expected, the folding of both the OmpA^L98R, V100R and EspPΔ5'^L1119R mutants was severely impaired (Fig. 6c, lanes 3 and 6; Supplementary Fig. 10a).

To examine our hypothesis further, we next examined the effect of darobactin on TAM-mediated OMP assembly. Darobactin is a naturally-occurring cyclic peptide that blocks BAM activity in vivo and in vitro by acting as a competitive inhibitor of β signal binding[14,36,37,91]. A cryo-EM study showed that darobactin binds to BamA(β1) through 6 backbone H-bonds and 1 side-chain H-bond (darobactin N2-BamA N427) (Fig. 6d, left panel)[91]. Strikingly, a structural prediction using Chimera and COOT suggests that TamA(β1) potentially binds to darobactin in the same fashion, through 7 H-bonds including 1 side-chain bond (darobactin N2-TamA E269) (Fig. 6d, right panel). In our OMP assembly assays, we first mixed BAM/PLE and TAM/PLE proteoliposomes with various concentrations of darobactin that were up to 100-fold higher than the concentration of TAM or BAM in a Tris buffer and incubated the samples at 30 °C for 5 min. We then added urea denatured OmpA or EspPΔ5' to the mixture, and the samples were incubated at 30 °C for another 15 min before being analyzed by Western blot. Interestingly, both TAM- and BAM-mediated OMP folding was significantly inhibited by darobactin in a concentration-dependent

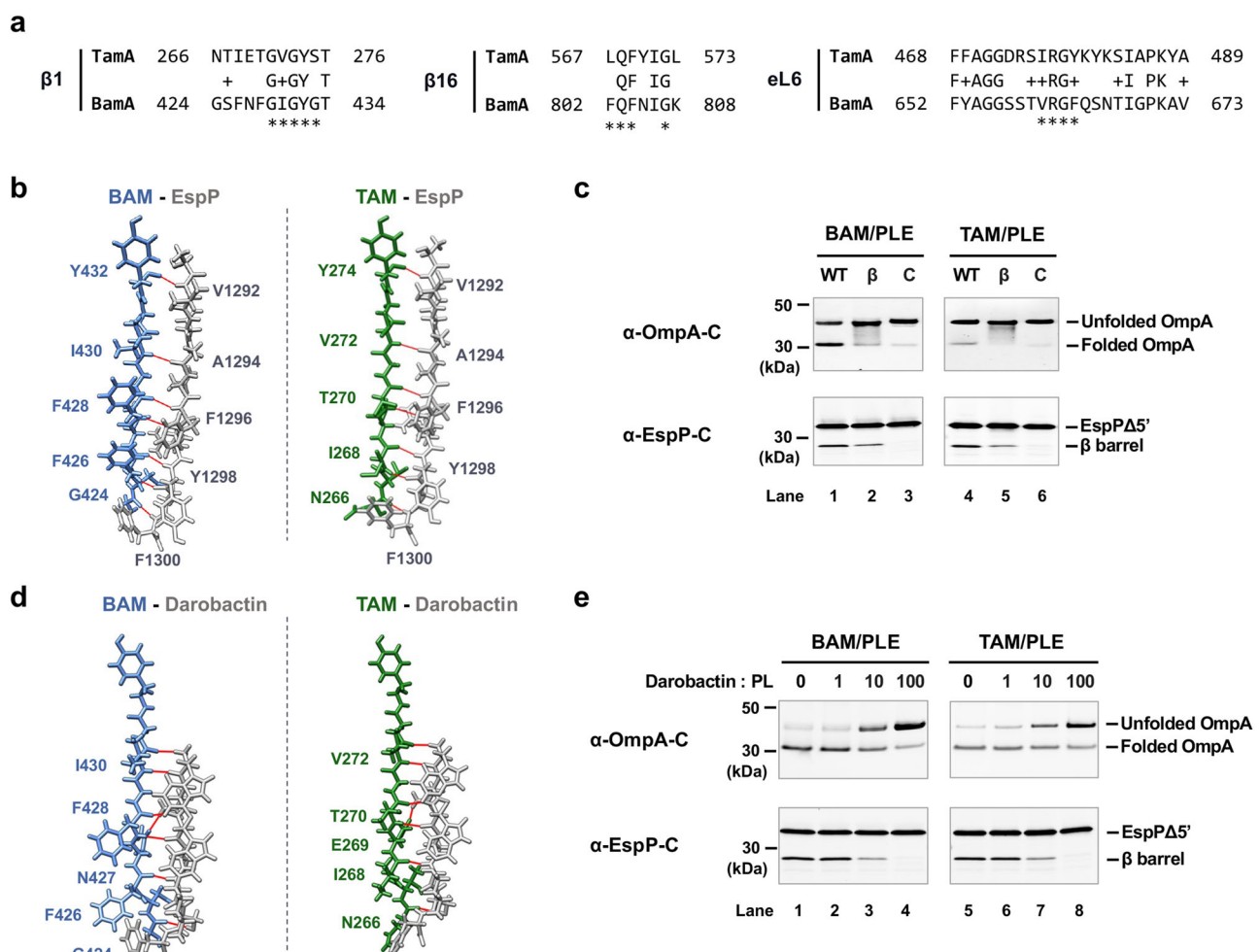

**Fig. 6 | TAM-catalyzed OMP assembly requires an interaction between TamA and the β signal of the substrate. a** The amino acid sequences of *E. coli* K-12 TamA and BamA β strands 1 and 16 (β1 and β16) and extracellular loop 6 (eL6) aligned based on the crystal structures of TamA (PDB: 4C00[42]) and BamA (PDB: 5D0O[24]). The BamA motifs conserved in Proteobacteria are indicated by asterisks[86]. **b** The binding of TamA(β1) to EspP(β12) (right) was predicted based on the BamA(β1)-EspP(β12) structural model (PDB: 7TTC[34]) (left). Seven H-bonds were predicted between TamA(β1) and EspP(β12) (right) and are denoted by solid red lines. Two H-bonds are predicted between TamA residues Tyr[274] and Asn[266] and EspP [TamA(Y274)-EspP(A1291) and TamA(Y274)-EspP(V1292); TamA(N266)-EspP(C1299) and TamA(N266)-EspP(F1300)], but only the closer H-bonds are shown. **c** Urea denatured wild-type (WT) OmpA or EspPΔ5′, β signal mutants (β) (OmpA[Y189A, F191A] or EspPΔ5′[Y1298A, F1300A]), or folding deficient controls (C) (OmpA[L98R, V100R] or EspPΔ5′[I119R]) were incubated with 2 μM BAM/PLE or TAM/PLE proteoliposomes at 30 °C for 2 min (for OmpA) or 15 min (for EspPΔ5′). Unheated OmpA or

heated EspPΔ5′ samples were subjected to SDS-PAGE, and assembly was monitored by Western blot using the appropriate anti-C terminal peptide antiserum. A representative experiment is shown. Three independent experiments were performed, and the other two are shown in Supplementary Fig. 10a. **d** The binding of darobactin to TamA(β1) (right) was predicted based on the structure of darobactin-bound BAM (PDG: 7NRI) (left)[91]. The seven H-bonds predicted between darobactin and TamA(β1) are denoted by solid red lines. **e** 2 μM BAM or TAM proteoliposomes were incubated with darobactin at 30 °C for 5 min. The amount of darobactin added to the reaction was varied to achieve the indicated darobactin-to-proteoliposome (PL) ratios. Urea-denatured OmpA or EspPΔ5′ was added to the reaction and incubated at 30 °C for 15 min. Unheated OmpA or heated EspPΔ5′ samples were subjected to SDS-PAGE, and assembly was monitored by Western blot using the appropriate anti-C terminal peptide antiserum. A representative experiment is shown. Three independent experiments were performed and a statistical analysis is shown in Supplementary Fig. 10b.

manner (Fig. 6e; Supplementary Fig. 10b). Consistent with previous results[35,92], we found that a linearized form of darobactin did not inhibit the activity of either BAM or TAM (Supplementary Fig. 10c). Taken together, our results provide strong evidence that TamA, like BamA, must bind to the β signal of client proteins to promote their assembly.

## Discussion

In this study, we obtained direct evidence that *E. coli* TAM can catalyze OMP assembly by analyzing its activity in vitro. Initially, we optimized our protocol to purify active TAM and found that, unlike BAM, TAM is sensitive to purification conditions and should be handled carefully. By reconstituting purified TAM into proteoliposomes, we could control the reaction parameters, such as the presence of chaperones and

inhibitors. We could also avoid the potential secondary effects of previous studies on TAM function that were conducted in vivo using depletion, mutant, or knock-out strains[38,39,43,51,93]. For example, disrupting *tamA* might subtly alter the structure of the OM and affect the assembly of a BAM substrate. Likewise, because BAM is likely required for the assembly of TamA, the depletion of BamA might mask the effects of depleting TamA[94]. Nevertheless, our observation that TAM catalyzes Ag43-β assembly in vitro is consistent with evidence that TAM promotes the efficient biogenesis of Ag43 in vivo and that Ag43 initiates dynamic movements in TAM in a reconstituted system[44,46]. The finding that three other OMPs were also assembled by TAM strongly suggest that it assembles other proteins in vivo (see below). Overall our data support a model in which TAM catalyzes OMP folding independently and is not only a co-factor that acts together with BAM.

As a corollary, we also found that while TamA is sufficient to catalyze a low level of OMP assembly, both TamA and TamB are required for efficient OMP assembly. Our data is consistent with the results of a previous in vivo study that showed that in the absence of TamA, but not TamB, the biogenesis of the TAM substrate p1121 is completely abolished[38]. The observation that OMPs assembled by TamA and TAM behaved similarly in heat-modifiability and PK-resistance tests despite the difference in folding efficiency indicates that TamB greatly facilitates the reaction catalyzed by TamA, but does not promote OMP assembly by a different mechanism. Although its function is unclear, TamB has been demonstrated to span the periplasmic space and interact with the N-terminal TamA POTRA domain[38,44,46,47]. Crystallographic studies show that the C-terminal 175 residues of TamB fold into a β-taco structure with a highly hydrophobic interior that can accommodate a single amphipathic β-strand[65], and AlphaFold2 and molecular dynamic (MD) simulations predict that the rest of the protein (except the N terminus) folds into a wider, somewhat twisted β-taco structure (Fig. 1b)[48]. Structural and biophysical studies also indicate that TamB applies force to TamA to alter its conformation and regulate its activity[44,46].

Interestingly, our results strongly suggest that TAM and BAM catalyze OMP assembly by similar mechanisms, at least in an in vitro assay that uses components purified by the same protocol. Consistent with this notion, the efficiency with which TAM and BAM catalyzed the assembly of four different OMPs was comparable over a wide range of proteoliposome concentrations. Furthermore, the backbones of *E. coli* TamA and BamA β-strands are structurally very similar[25,26,42,47], and in both proteins the β strand that binds to the β signal of incoming OMPs (β1) and the C-terminal β strand that together with β1 forms the β barrel seam (β16) share sequence motifs that are conserved in TamA and BamA Proteobacterial homologs. We found that *E. coli* TamA(β1) is not only predicted to form backbone H-bonds with the EspP β signal that parallel the H-bonds formed by BamA(β1) (Fig. 6b), but that key mutations in the OmpA and EspPΔ5′ β signal inhibit the assembly mediated by TAM (Fig. 6c). This is an especially important observation given that several lines of evidence show that a stable BamA(β1)-β signal interaction initiates OMP assembly[32,34,35,37]. In addition, consistent with our prediction that a competitive inhibitor of β signal binding (darobactin)[91] would bind to TamA(β1) mainly through backbone H-bonds as it binds to BamA(β1), we found that darobactin inhibited TAM-mediated OMP assembly (Fig. 6d, e). Taken together, the results suggest that TamA and BamA both catalyze OMP assembly by opening laterally and forming asymmetrical hybrid barrels with client proteins. In should be noted that we also observed a few differences between BAM-mediated and TAM-mediated OMP assembly that raise the possibility that the two reactions proceed by slightly different mechanisms. First, the ratio of the four folded forms of Ag43-β and their sensitivity to PK digestion depended on the factor that catalyzed Ag43-β assembly (Fig. 3c). It is difficult to determine, however, if these disparities are due to a slight difference in the mechanisms of assembly, or if there are subtle structural differences in the TAM/PLE and BAM/PLE proteoliposomes that affect folding or the accessibility of the protease. In addition PLE, which creates a more physiological environment than synthetic lipids, promoted the highest level of TAM activity but was previously observed to promote a relatively low level of BAM activity, and while TAM catalyzed the assembly of OmpA more rapidly than EspPΔ5′, BAM previously catalyzed the assembly of EspPΔ5′ more rapidly than OmpA[73]. It is unclear, however, if these disparities simply reflect the use of slightly different protocols to purify BAM and TAM.

It should be emphasized that taken together with previously published experimental and bioinformatic results, our results provide a proof-of-concept that TAM can function as an independent β barrel insertase, but do not establish exactly how environmental or growth conditions might affect its role in the physiology of different organisms or its range of substrates. In this regard, it is noteworthy that the assembly of EspP (an autotransporter like Ag43, but a member of a different subfamily) in vivo has been reported to be unaffected by the disruption of *tam* in *E. coli* [51]. This study, however, was only conducted in a monoculture in a minimal medium at 37 °C and does not rule out the possibility that TAM catalyzes EspP assembly under different conditions[51]. Furthermore, although our data strongly suggest that *E. coli* BAM and TAM are functionally redundant, a limitation of our study is that we could not determine whether TAM can replace BAM in vivo under standard laboratory conditions (in which BAM is essential and is 20 - 40 times more abundant than TAM[49,50]) because BamA depletion experiments generated ambiguous results. Nevertheless, the finding that *tam* deletions or mutations can disrupt the stress resistance, colonization, and virulence of specific bacteria[38,43,44,46,47,52,53,59-63] supports the idea that the role of TAM in OMP assembly is condition- and organism-dependent.

Based on our in vitro results, previous genetic, structural and computational studies, and studies on the interactions between the two TAM subunits, we propose a model in which a segment of TamB binds to the first POTRA domain of TamA while its C terminus binds to TamA β1 and thereby serves as a placeholder for incoming OMPs (Fig. 7). Indeed Sam50, the mitochondrial homolog of BamA, forms a dimer in which the second subunit appears to perform a placeholder function[95]. In our model a subset of preferred OMPs are initially bound by classical periplasmic chaperones (e.g., SurA) and then transferred to TamB. Subsequently TamB uses its β-taco structure to escort OMPs that have at least partially folded into an amphipathic β-sheet across the periplasm to the OM and applies a force to the TamA POTRA domains that releases its C terminus from the TamA β barrel seam[44,46,65]. TamA, like BamA, then promotes OMP assembly by forming an asymmetrical hybrid barrel with substrates. After the substrate is released from TamA via a stepwise strand exchange reaction that enables its first and last β-strands to form a closed barrel, TAM then resets to a ground state in which it is ready to receive a new substrate. Given that it is unclear whether OMPs bind to TamB, however, it is possible that chaperones target them directly to TamA, perhaps by interacting with a POTRA domain (as SurA binds to the first POTRA domain of BamA[96]), and thereby initiate the assembly cycle by triggering a conformational change in TamB. In any case, TAM seems to resemble BAM in that the core BAM subunit (BamA) is sufficient to catalyze OMP assembly (albeit very inefficiently) while the lipoproteins serve regulatory roles that are essential to maximize function[97,98]. It is conceivable, however, that the accessory proteins (rather than the core subunits) possess unique capabilities that explain why BAM is the primary OMP insertase in *E. coli* while TamB is the major BamA binding partner in very different organisms like *Borrelia*[40,66].

In light of our finding that TAM can catalyze OMP assembly, it is unclear how to interpret genetic and bioinformatic evidence that strongly suggest that TAM plays a key role in phospholipid homeostasis. It has been shown that deleting both *tamB* and *yhdP*, a gene that encodes a protein that affects phospholipid transport from the IM to the OM[99], causes pleiotropic defects including increased OM permeability and cell lysis, and that the deletion of *tamA* and *yhdP* produces the same phenotype as the deletion of *tamB* and *yhdP*[67,68]. Furthermore, the finding that the assembly of at least OmpA and OmpC is not affected in a ΔtamBΔyhdP strain but that the loss of OM integrity can be suppressed by mutations that prevent the removal of phospholipids from the OM strongly suggests that the phenotype is caused by defects in anterograde phospholipid transport[67]. Interestingly, TamB and YhdP are both in the AsmA family of proteins and are nearly identical in size and predicted structure[67]. It should be noted, however, that because there is no direct evidence that TamB binds to phospholipids and defects in lipid transport that were observed in a BamA depletion strain in an early study[100] were likely an indirect effect of OMP assembly defects, the results of these genetic experiments

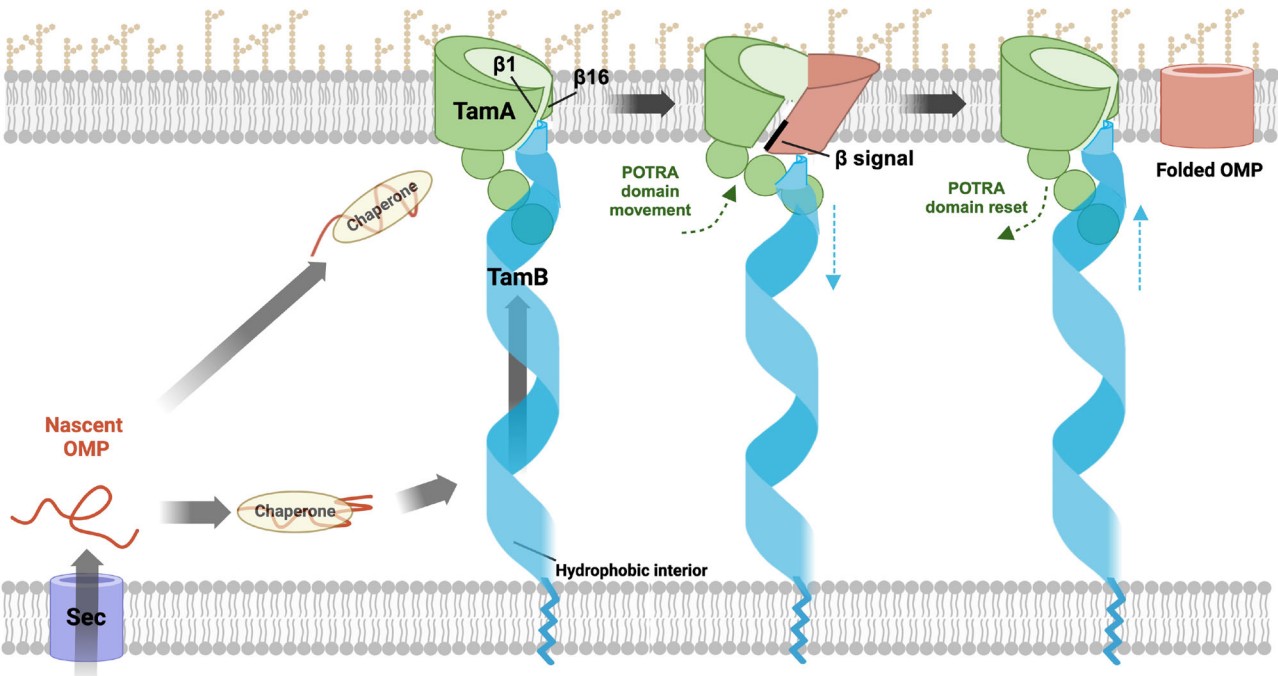

**Fig. 7 | Model of OMP assembly by TAM.** Nascent OMPs are transported into the periplasm through the Sec complex and bind to periplasmic chaperones (e.g., SurA) which maintain them in an assembly competent conformation. Possibly due to partial folding and the exposure of a hydrophobic surface, a subset of OMPs might subsequently bind to TamB, which could then use the hydrophobic interior of its β-taco structure to escort them to TamA. Alternatively, chaperones might target specific OMPs directly to TamA and thereby trigger conformational changes in TamB. In either scenario TamB applies a force to the TamA POTRA domains that alters their position and thereby promotes the release of the C-terminus of TamB from TamA β1. Like BamA, TamA subsequently forms a hybrid barrel with OMPs through a strong interaction between β1 and the β signal of the substrate. After the substrate curves inward to form a barrel-like structure, its first and last β strands form a closed seam via a strand exchange reaction and the fully folded OMP is released into the lipid bilayer. Conformational changes in the TamA POTRA domains and TamB that occur during the OMP assembly cycle are denoted by green and blue arrows, respectively. It should be noted that available evidence suggests that TamB also mediates the anterograde transport of phospholipids, presumably through the interior of the β-taco. It is conceivable that the lipids are transported in association with OMPs. This figure was created with BioRender.com, released under a Creative Commons Attribution-NonCommercial-NoDerivs 4.0 International license.

should be interpreted cautiously. Nevertheless, one possible explanation for the apparent dual function of TAM is that TamB is a somewhat generic transporter of hydrophobic molecules, which might include a subset of OMPs that are partially folded in the periplasm and expose a hydrophobic exterior. In the case of lipid transport, TamA is required to anchor TamB and ensure that the lipids reach their target membrane. It is unclear, however, if the TamB C-terminus, which has been predicted to bind to β1 of the open TamA β barrel[48], would be released during lipid transport or if the putative TamA β barrel-TamB interaction is important only to facilitate a transition between two different functional stages of OMP assembly. A second possibility is that partially folded OMPs "piggyback" on phospholipids (or vice-versa) and both types of molecules reach the OM together. Finally, in light of recent evidence that TamB and YhdP primarily transport distinct classes of lipids and that the phospholipid composition of the OM is regulated by changing the relative activity of each protein[101], it is also possible that TAM transports OMPs when it is not needed for lipid transport.

## Methods

### Bacterial strains, antibiotics, and antisera

*E. coli* strain BL21(DE3) (Thermo Scientific, catalog number EC0114) and BL21-CodonPlus (DE3)-RIPL (Agilent, catalog number 230280) were used in this study. Cells were grown in LB supplemented with ampicillin (100 μg/ml) or kanamycin (30 μg/ml) as needed. Rabbit polyclonal antisera were generated against HPLC-purified peptides derived from the TamA N terminus (NH2-NVRLQVEGLSGQLEKNV-RAQC-COOH), the TamA C terminus (NH2-CPVADKDEHGLQFYIGLGPE-COOH), and the FadL C terminus (NH2-CHGQSVKINEGPYQFESEGK-COOH). Rabbit polyclonal antisera generated against BamA and BamD C-terminal peptides, a peptide derived from OmpA extracellular loop 4, an OmpA C-terminal peptide, an EspP C-terminal peptide, and the Ag43 β-domain have been described[12,18,72,73,102]. In general, rabbit antisera were used at a dilution of 1:5000-1:10000. The mouse monoclonal anti-His tag antibody was obtained from Genscript (catalog number A00186) and used at the dilution recommended by the manufacturer.

### Plasmid construction

All plasmids used in this study are listed in Supplementary Table 3. To construct pXW47(P$_{trc}$-(8xHis)-TamAB), the DNA fragment encoding TAM was amplified with an additional 35 bp upstream fragment (*tamAB$_{-35-5510}$*) by PCR using *E. coli* MC4100 genomic DNA and the primer pair XW63/XW64 (oligonucleotide primers used in this study are listed in Supplementary Table 4). Using *tamAB$_{-35-5510}$* as the template, a DNA fragment (*tamAB$_{-35-66}$-His$_8$*) carrying the ribosomal binding site and encoding the TamA signal peptide followed with an octa-histidine tag (NH2-HHHHHHHHGGSGGSGG-COOH) were amplified using the primer pair XW65/XW66. The DNA fragment (*His$_8$-tamAB$_{67-5510}$*) encoding the octa-histidine tag and the mature region of TamAB was amplified using primers XW67 and XW68. The PCR fragments *tamAB$_{-35-66}$-His$_8$* and *His$_8$-tamAB$_{67-5510}$* were then assembled into pTrc99a by Gibson assembly (New England Biolabs catalog number E2611S)[103]. To generate pXW48 (P$_{trc}$-(8xHis)-TamAB-TamB), the DNA fragment encoding the second copy of TamB was amplified using *tamAB$_{-35-5510}$* as the template and the primer pair XW70/XW71. The resulting PCR product was assembled into pXW47 using gBlocks XW72

and XW73 by Gibson assembly[103]. To construct pXW49 (P$_{trc}$-(8xHis)-TamA), the DNA fragment encoding His-tagged TamA was amplified using pXW47 as a template and primers XW65 and XW69, and was inserted into the vector pTrc99a by Gibson assembly[103]. To construct plasmid pXW50 (P$_{trc}$-(8xHis)-TamA$^{G271C, G574C}$-TamB), TamA G271C and G574C substitutions were introduced into pXW48 using the Quik-Change Site-Directed Mutagenesis Kit (Agilent catalog number 200524). Plasmids that encode *bamABCDE$_{8His}$-bamB* (pYG120) and *ompA, espPΔ5'* and *ag43* and their derivatives used to produce substrates for OMP assembly assays have been described[12,18,24,70,73]. A plasmid that encodes *fadL* (pET303::*fadL26-446)* was kindly provided by Dr. Joanna Slusky.

## Expression and purification of TAM and BAM

To produce TAM and TamA, BL21-CodonPlus(DE3)-RIPL transformed with pXW48 or pXW49 were grown at 37 °C overnight in 2xYT medium (Sigma-Aldrich, catalog number Y2377) containing 100 µg/ml ampicillin. The cells were washed and diluted 1:100 into 12 L 2xYT medium and grown at 37 °C to OD$_{600}$ = 0.4–0.5. (Inducing TAM expression at a higher OD$_{600}$ dramatically reduces the yield of the complex). Cultures were chilled on ice and 50 µM IPTG was added to induce protein expression (a relatively low IPTG concentration helps to relieve the toxicity of TAM overexpression). To produce BAM, BL21-Codon-Plus(DE3)-RIPL transformed with pYG120 were grown at 37 °C in 2 L 2xYT medium to OD$_{600}$ = 0.6–0.8. Cultures were chilled on ice and 400 µM IPTG was added to induce protein expression. After 16–18 h incubation at 16 °C, cells were harvested by centrifugation (4000 × g, 20 min, 4 °C), washed with cold PBS, pH 7.4, flash-frozen and stored at −80 °C. Samples were collected before and after IPTG induction and analyzed by Western blot using an anti-His antibody to confirm protein expression.

Cells were lysed at 5 °C using one pass through a continuous flow cell disruptor (Constant Systems BT40) at 30,000 p.s.i. The cell lysates were centrifuged twice to pellet unbroken cells (4000 × g, 15 min, 4 °C; then 6000 × g, 20 min, 4 °C). The supernatants were centrifuged in a Beckman Type 70 Ti rotor (311,000 x g, 65 min, 4 °C) to pellet the membranes. The resulting pellets were resuspended in Solubilization Buffer (PBS, 1% n-dodecyl-β-maltoside (DDM, Anatrace, catalog number D310), 37 mM imidazole, pH 7.4 at a concentration of 8–12 mL per liter of culture) and dispersed by a Dounce homogenizer. The resuspended membranes were rotated at 15 rpm at 4 °C overnight. The insoluble material was pelleted in a Beckman Type 70 Ti rotor (257000 x g, 30 min, 4 °C). The supernatant was loading onto 5 mL Ni-NTA resin (Qiagen, catalog number 30230) equilibrated with buffer A (PBS, 0.1% DDM, 37 mM imidazole, pH 7.4). After the supernatant was slowly passed over the resin three times, the resin was washed with 150 mL buffer A. The target protein was eluted with buffer B (PBS, 0.1% DDM, 500 mM imidazole, pH 7.4) in eighteen 1.5 mL fractions. SDS-PAGE was performed to identify the fractions that contained the protein(s) of interest. Typically 10–18 fractions were pooled and passed through Zeba™ spin desalting columns (Thermo Fisher; 7 K MWCO, 10 mL for TamA, catalog number 89893; 40 K MWCO, 10 mL for TAM or BAM, catalog number A57765) to exchange the buffer with buffer C (20 mM Tris, pH 8.0, 0.03% DDM). Proteins were concentrated using Amicon® Ultra-15 centrifugal filters (Millipore Sigma; 10 K MWCO for TamA; 30 K MWCO for TAM or BAM, catalog numbers UFC901008 and UFC903008). The protein concentrations were estimated by measuring $A_{280}$ ($\varepsilon_{TamA}$ = 114,250 m$^{-1}$ cm$^{-1}$; $\varepsilon_{TamAB}$ = 249,830 m$^{-1}$ cm$^{-1}$; $\varepsilon_{Bam}$ = 294,630 m$^{-1}$ cm$^{-1}$)[104]. The proteins were diluted to 20 µM by 20 mM Tris, pH 8.0 and clarified by centrifugation (12,000 × g, 5 min, 4 °C) before reconstitution.

## Reconstitution of TAM and BAM into liposomes

The chloroform solutions of *E. coli* polar lipid extracts and synthetic phospholipids (Avanti Polar Lipids, catalog numbers 100600 C,

850457 C, 850355 C, 850345 C, 850335 C) were transferred into glass tubes, and the organic solvent was evaporated with a nitrogen stream. The lipids were then placed in a vacuum desiccator overnight and rehydrated to 8 mg/ml in 20 mM Tris, pH 8.0. The lipid solutions were incubated at 42 °C for 30–60 min with occasional vortexing and then sonicated for 10–30 min in an ultrasonic water bath (Branson 1800). Subsequently the lipids were clarified in a second sonication step using an ultrasonic processor at 40% of the maximum amplitude (Cole Parmer model CPX130PB). Aggregates were removed by centrifugation (4000 × g, 5 min, room temperature). To generate liposomes, the lipids were extruded using an Avanti mini-extruder with 100 nm filters (Avanti Polar Lipids, catalog number 610000-1EA) and clarified by centrifugation (12,000 × g, 5 min, 4 °C).

For reconstitution, 1.25 mL of 20 µM TAM or BAM was incubated with 625 µL liposomes on ice for 5 min. The mixture was diluted to 25 mL with 20 mM Tris, pH 8.0, rotated at 15 rpm at 4 °C for 45 min, and centrifuged (4000 × g, 10 min, 4 °C) to remove aggregates. The reconstituted proteoliposomes were pelleted in a Beckman Type 70 Ti rotor (311,000 × g, 95 min, 4 °C) and resuspended in 200–500 µL 20 mM Tris, pH 8.0. The resulting proteoliposomes were divided into 10–50 µL aliquots, flash-frozen and stored at −80 °C. The final protein concentration was generally determined by the Bio-Rad DC protein assay (catalog number 5000112). (If the protein yield was low, the protein concentration was determined by quantifying the sample concentration using a TAM or BAM sample of known concentration as a standard. Both samples and standards were resolved on SDS-PAGE, visualized by Western blot, and analyzed by ImageJ.) To determine reconstitution efficiency, samples were collected before and after reconstitution and resolved by SDS-PAGE and Coomassie Blue staining. The orientation of TAM and BAM in proteoliposomes was determined by trypsin digestion as described[70].

## OMP assembly assays and data analysis

Wild-type and mutant forms of OmpA, EspPΔ5', Ag43-β, and FadL were expressed in BL21(DE3) transformed with the appropriate plasmid, cells were lysed with BugBuster solution (Millipore Sigma, catalog number 71456), and the proteins were purified without protease inhibitors as previously described[18]. Because OmpA that was purified by this method and added to assembly reactions that contained TamA/PLE was partially degraded during the incubation, OmpA was purified by a slightly different method[73] for all TamA-mediated assembly reactions. Purified proteins were diluted to a concentration of 6 µM in 8 M urea, 20 mM Tris, pH 8.0, flash frozen, and stored at −80 °C. SurA and DegP$^{S210A}$ were purified as previously described[83]. Skp was obtained from MyBioSource.com. OsmY was kindly provided by Dr. Zhen Yan and Dr. James Bardwell.

OMP folding assays were performed by slightly modifying a previously described protocol[18,70,73]. TamA/PLE or TAM/PLE proteoliposomes were sonicated in a Branson water bath for 5–7 min at room temperature and BAM/PLE proteoliposomes were sonicated for 7–15 min. A urea denatured OMP (0.2 µM) and 0.1–2 µM proteoliposomes were immediately mixed in 20 mM Tris, pH 8.0 after sonication. In some experiments, a chaperone (2 µM) was added as indicated. An additional 2–3 min of sonication would be needed if the proteoliposomes were not used immediately. Reactions were incubated at 30 °C for up to 1 h with 600 rpm shaking and stopped by adding 4x NuPAGE LDS loading buffer (Thermo Fisher, catalog number #P0007) for OmpA or 2x SDS loading buffer (Quality Biological, catalog number 351-082-661) for EspPΔ5', Ag43-β, and FadL and placing the tubes on ice. In some experiments, half of each sample was kept on ice, while the other half was heated at 95 °C (15 min for OmpA, 10 min for EspPΔ5', Ag43-β, and FadL). For PK digestions, 5 µg/mL PK (Roche, catalog number 03115852001) was added to the reactions and kept on ice for 10 min. The digestions were stopped by adding 8 mM PMSF and loading buffer. The OmpA

samples were resolved on 12% Bis-Tris minigels (Thermo Fisher, catalog number NP0343BOX) using MES buffer and assembly was determined by Western blotting. For blots containing PK-treated OmpA samples, an antiserum raised against an OmpA extracellular loop 4 peptide was used. Otherwise, an antiserum generated against an OmpA C-terminal peptide was used. The EspPΔ5′, Ag43-β, and FadL samples were resolved on 8–16% Tris-glycine minigels (Thermo Fisher, catalog number XP08165BOX), and assembly was determined by Western blotting using an appropriate antiserum and the IR Dye 680LT goat anti-rabbit secondary antibody (LI-COR, catalog number 926-68021). Proteins were visualized using an Amersham Typhoon imager (Cytiva) at an excitation wavelength of 685 nm. To determine the percent of OMPs that were assembled, Western blots were analyzed using ImageJ v 1.54 software as previously described[73].

To generate kinetic data for OmpA and EspPΔ5′ assembly, assembly was monitored by removing samples from reactions at various time points between 0–60 min and calculating the fraction of OmpA that migrated rapidly in the absence of heat or the fraction of EspPΔ5′ that had undergone proteolytic processing. Four to five independent experiments were performed. Data was fit to a single exponential model, and the rate constant (K) and the time required to reach 50% maximal folding ($t_{1/2}$) were calculated using GraphPad Prism 8.

### Darobactin inhibition assay
Native darobactin (kindly provided by Kim Lewis) or a linear form of the cyclic peptide (0–200 μM) and sonicated proteoliposomes (2 μM) were mixed in 20 mM Tris, pH 8.0 and incubated at 30 °C for 5 min with shaking (600 rpm). A urea denatured OMP (0.2 μM) was then added and incubated for an additional 15 min. OMP assembly was determined by Western blot as described above.

### Structural alignments and predictions
β-barrel superimposition of the crystal structure of TamA (PDB: 4C00[42]) and the cryoEM structure of BamA bound to darobactin (PDB: 8BVQ[92]) was generated using UCSF Chimera 1.15[105]. RMSDs between the TamA and BamA β barrels were calculated using Pymol 2.5.5 To examine potential interactions between TamA(β1) and the substrate EspP, the TamA crystal structure (PDBID: 4C00) was modelled onto the BamA backbone of the cryoEM structure of BamA in complex with EspP (PDB: 7TTC[34]). Investigation of possible interactions between TamA β1 and darobactin utilized the TamA crystal structure (PDBID: 4C00) modelled onto the BamA backbone of the crystal structure of BamA bound to darobactin (PDBID: 7NRI[91]). Models were first aligned in UCSF Chimera 1.15, and then refined using COOT 0.9.0[105,106]. Models were evaluated using MolProbity[107]. H-bonds were inspected using UCSF ChimeraX 1.2.5[108].

### Reporting summary
Further information on research design is available in the Nature Portfolio Reporting Summary linked to this article.

## Data availability
The protein structures that we refer to can be accessed through the PDB using the following URLs: 4C00, 7NRI, 8BVQ, 7TTC, 5D0O, 1G90, 2MQE, 3SLO, 1T16. Source data are provided with this paper.

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

## Acknowledgements

We thank James Bardwell, Kim Lewis, Joanna Slusky, and Zhen Yan for providing reagents and Matt Doyle for providing insightful comments on the manuscript. We would also like to thank Jenny Hinshaw for her help with model building. This work was supported by the Intramural Research Program of the National Institute of Diabetes and Digestive and Kidney Diseases.

## Author contributions

X.W. and H.D.B. designed the research, X.W. performed the experiments, S.B.N. performed the structure alignments, X.W. and H.D.B. analyzed the data, and X.W. and H.D.B. wrote the paper.

## Competing interests

The authors declare no competing interests.
