## [Peer Review File · Nature Communications]

The translocation assembly module (TAM) catalyzes the assembly of bacterial outer membrane proteins in vitroReviewer #1 (Remarks to the Author):

Remarks to Authors,

The manuscript by Wang et al reports the TAM function as independent OMPs insertases as well as BAM by establishing the specific *in vitro* reconstitution assay to analyze TAM functions.

Unfortunately, however, I can not support the publication of this manuscript in its current form, because I do not think that the data establish the TAM contributed to the OMPs assembly into the proteoliposomes well and I am therefore not convinced of the main conclusion of the work.

Most strikingly in my eyes is the insufficiency of purity of the TAM. Fig. 2d shows that not only TamA and TamB but also multiple proteins were identified in the TAM proteoliposomes, including bands with similar molecular weights to BamA and BamD (see attached image). Since these bands were not identified in the TamA proteoliposome (Fig. S2C), it is possible that these proteins were contaminated with the addition of TamB. Previous reports have demonstrated that even the presence of BamA or BamD alone in liposomes has OMP assembly activity for those proteoliposomes (Hagan et al., 2010 Science). The worst-case scenario, therefore, is that the addition of TamB has led to the contamination of BamA or BamD or both, resulting in OMP assembly activity. However, the authors have not been able to provide sufficient data to dispense with this worst-case scenario. To dispel this possibility, the authors should show the absence of the insertase activity in assembly experiments with proteoliposomes prepared using a loss of function mutant of TAM.

The authors propose in Fig. 7 that TamB interaction with the substrate facilitates TAM-mediated assembly. However, no data are presented in this paper that indicates a direct interaction between TamB and the substrate. Because previous studies have shown that TamB stimulates conformational changes in TamA (Shen et al., 2014 Nat Commun), the reasons why TamB enhances the insertase activity of TamA are not limited to the molecular mechanism proposed by the authors.

Since Nature Communication is a general journal targeting a broad readership, I believe that confirmation under physiological conditions as well as validation in *in vitro* assays is essential. If the authors claim that TAM has the same insertase function as BAM, the authors should provide results of experiments showing that the effects of loss of function mutants of the BAM can be suppressed by overexpression of the TAM.

Reviewer #1 Attachment on the following page

Coomassie blue staining

This is a high-contrast image of Fig. 2d. The multiple proteins were observed. Especially red and blue arrows corresponds the similar molecular weight as BamA and BamD, respectively.

C

(C)

They are high-contrast images of Fig. 2C (left) and Fig. S2C (right). They are the same condition except for samples, left; TamA and TamB, right; TamA only. TAM containing proteoliposome contained the proteins indicated by arrowheads, but not TamA proteoliposome.

Reviewer #2 (Remarks to the Author):

Gram-negative bacteria have an outer membrane outside of their peptidoglycan layer that provides an essential layer of protection against environmental toxins and insults. However, this membrane is far removed from the bacterial cell's protein and lipid biosynthetic machinery and therefore the biogenesis of the outer membrane is a challenging cellular process. The beta-barrel assembly machinery is responsible for the assembly of the vast majority of outer membrane proteins (OMPs) and is composed from a core BAM complex and a translocation and assembly module (TAM) that together ensure that all beta-barrel proteins are assembled with optimal efficiencies.

Over the past 10 years, a great deal of work has been invested into understanding these assembly machines, with the BAM complex having the most attention. In the case of the TAM it is clear from kinetic studies conducted *in vivo* that the TAM adds to the speed and efficiency of a number of OMPs. The TAM has been purified and reconstituted into membranes to demonstrate that the TAM can mediate the insertion of an OMP into the membrane layer (for example, measured by neutron reflectometry), but it has never been demonstrated whether the purified TAM can mediate the folding of OMPs into their final, native form. In this manuscript by Wang et al, that proof is finally provided and the assay system created for that purpose is used to monitor the folding of 4 OMPs, each with distinct extracellular domains in addition to their membrane-integral beta-barrel domain. This is a powerful assay and it is used to great effect. The quality of the data is excellent and I have no suggestions for additional wet-lab experimental work. The major short-coming of the paper is considerable inaccuracy when it comes to crediting past work published by other research teams and an inaccurate over-emphasis on both what was missing in our knowledge and in seeking to propose controversies and complications that do not exist in the literature. The specific suggestions for addressing these short-comings, some minor and some major, are detailed below.

1. Title: please replace the name "Tam complex", which is not used in the literature, with the accepted name "TAM" or the long form Translocation Assembly Module. This also provides for internal consistency as TAM is used elsewhere in the paper (other than the legend of Figure 1).
2. Abstract (lines 29-30): Please correct the statement "Although its function is controversial, TAM has been proposed to play a critical role in the assembly of a small subset of *E. coli* OMPs based primarily on experiments conducted *in vivo* using *tamA* and/or *tamB* deletion or mutant strains". This statement is not factual. There is no controversy about the function of the TAM with published evidence from many papers showing that it functions in the assembly of several types of OMPs (autotransporters, inverse autotransporters, fimbrial ushers and efflux pumps; not all of this evidence is cited in the current manuscript). This evidence has included experimentation both *in vivo* - using *tamA* and/or *tamB* deletion or mutant strains - as well as *in vitro* reconstitution experiments using quartz crystal microbalance measurements and/or analysis by neutron reflectometry.
3. Introduction (lines 120-121) Please correct the statement "To complicate matters, however, recent evidence strongly suggests that TamB is a member of the AsmA-like family of proteins and works with TamA to promote phospholipid transport to and from the OM of *E. coli*[59,60]." That the TAM might also play a role in phospholipid transport in the absence of the major phospholipid transporter (YhdP) is neither a complication nor controversial. That TamB is a member of the AsmA-superfamily of proteins is not a new observation, and both References [59 and 60] cite (PMID: 25994932) as the source of these observations and for the characterization

of the AsmA-related groups of proteins including YhdP, TamB, and YdbH.

4. (Lines 124-126, as per point 1): In order to correctly acknowledge the previous works on TamB, it would be important to refine the text “To overcome the potential problem of indirect effects of TAM mutations in studies performed in vivo and to test the ability of TAM to catalyze OMP assembly directly, we developed a novel method to purify and reconstitute TAM into lipid vesicles in vitro”.

A more accurate statement of the contribution would be to recognize that the current study is the first to show that the TAM is necessary and sufficient to fold the beta-barrel domains of four OMPs in a purified system, without the assistance of the BAM complex.

5. Figure 1. (lines 143-147): “In superimposing the published structures of the E. coli BamA and TamA β barrels^{44,61}, we noticed a remarkable degree of overlap in the two β barrel backbones that has been previously reported⁶² (Fig. 1A). Although most of the loops in the TamA β barrel are shorter than their cognate loops in the BamA β barrel, the fundamental similarity in overall structure suggested a similarity of function.”

As noted in this text, these are published observations by other authors. Why is it being published again in this paper? This text (and Figure 1B) should be removed. Figure 1B is an excellent figure to start the paper.

6. Figure 1 legend: please replace the word “Tam complex or TAM” with the accepted name “TAM”. This provides for internal consistency as TAM is used elsewhere in the paper.

7. Figure 2. The data is excellent and convincing, but the protein bands are barely visible and the background is too white (i.e. the contrast on the images is too high, potentially obscuring other species in the gels). The figures should be adjusted to rectify this issue.

8. Figure 5, Figure 6. The primary data is excellent and convincing. I was not able to find any reference to how the “% cleaved” and “% folding” figures were calculated, and no errors are noted – is there no margin of error in these measurements (for example, multiple quantifications of the same data and/or normalization corrections on the multiple experiments)?

9. Figure 7. Why is TamB represented as a half-pipe emphasizing lipid transport here, and yet represented as a superhelical twist in Figure 1? The scale for these two figures is important, can they be adjusted so that the scale is (approximately) similar in the structural perspective (Figure 1) and the cartoon representation (Figure 7).

10. Figure 7. It should be made clear in the Legend or the main text that the binding of substrate protein into the beta-taco groove, as well as the presence of lipids in the beta-taco groove, are purely speculation. Neither of these events has been experimentally documented.

11. Supplementary Figure S1. In the absence of heat, the purified TAM migrates at a size somewhere between 480kDa and 720kDa. What is the basis for this size? This should be noted in the main text of the paper.

12. Supplementary Figure S1. In the presence of heat, TamA migrates at a size somewhere between 148kDa and 242kDa, despite being a protein of 50-60kDa in size. What is the basis for this size? This should be noted in the main text of the paper.

13. Supplementary Figure S3. I leave this to the discretion of the authors: could you consider making Fig S3A part of main Figure 3? I would have found this summary information helpful when reading the main text.

14. Supplementary Figure S6/Table S1. The reasoning for the selection of the species for which the sequence analysis is done is hard to reconcile. Table S1 notes that some of the species do or don't have all three Omp85 types in their genomes. Why not choose species which do have all three types for the analysis? Also, why is the analysis done with so few protein sequences? The motifs that are generated are highly biased as a result. In the current study, the size of the conserved letters here would suggest almost strict conservation across species. However, this

is not what is found in previously published work where a comprehensive analysis of all available sequences was used. In the previous unbiased analyses, relatively few residues are highly conserved, and none are absolutely conserved.

15. Discussion. In my opinion, the Discussion would be greatly improved by contrasting what is shown herein that focusses on the correct folding of OMPs mediated by the TAM (the novelty of this work) with the previous demonstration that the reconstituted TAM mediated the insertion of a model OMP (i.e. the equivalent of EspP) (PMID: 25341963 and PMID: 26243377).

Reviewer #3 (Remarks to the Author):

Review Response for The purified E. coli TAM complex catalyzes the assembly of bacterial outer membrane proteins in vitro

Wang et al. present the first example of OMP folding in vitro by the TAM complex. After expression and purification of the TAM complex (consisting of TamA and TamB), they demonstrate that TAM can fold four OMPs of varying sizes into proteoliposomes. They suggest that this is via a mechanism that is similar to the BAM assisted folding mechanism because addition of darobactin or Skp and mutation of the β -signal all reduce folding in a manner similar to their action on BAM.

This is a timely paper as the evidence for TAM function is unclear and based on indirect genetic evidence. This manuscript reflects that TAM can fold OMPs, but it remains unanswered as to whether this is the primary function of TAM and if TAM is involved in lipid homeostasis.

Some key points to address include:

- McDonnell et al. (bioRxiv 2023) have used Alphafold2 multimer to predict the complex between TamA and TamB. It suggests that TamB interacts with TamA at the 'lateral gate' by forming a beta augmented hybrid barrel. How then can TamA fold OMPs without first dissociating from TamB? Could the authors please comment on their model for TAM mediated folding in the light of this model structure? Do the authors have any evidence that TamA and TamB are still interacting in the beta signal variants or in the samples where darobactin has been added?
- The examination of the addition of specific chaperones with TAM and BAM is interesting, but is it not surprising that the addition of SurA to the BAM sample does not increase the folding of the substrate OMP given that previous papers have shown that SurA increases folding of OmpA variants by between 50-80% (Devlin et al 2023, Schiffrin et al 2022). The experiment on chaperones and TamAB was only performed with EspP. Could this be done with OmpA too?
- In Figure 1b when TAM is assembled in a proteoliposome, the TamB transmembrane helix does not appear to be assembled into a liposome, would this not be the case?
- The TamA structural 'predictions' are strange – simply aligning a beta strand next to another will create H-bonds between backbone atoms and does not indicate that it binds. Terming this in the methods as Model building and refinement suggests that there is experimental data to refine into which is not correct. Given the information available, structural alignments are only

possible and the ability to say that the binding of darobactin or substrate does not cause any major clashes. In figure 1, the alignments between TamA and BamA would benefit from RMSD values and would move the text away from terms such as the barrels are similar (line 433)

RESPONSE TO REVIEWERS' COMMENTS (all major changes are highlighted in the text)

Reviewer #1 (Remarks to the Author):

Remarks to Authors,

The manuscript by Wang et al reports the TAM function as independent OMPs invertases as well as BAM by establishing the specific *in vitro* reconstitution assay to analyze TAM functions. Unfortunately, however, I can not support the publication of this manuscript in its current form, because I do not think that the data establish the TAM contributed to the OMPs assembly into the proteoliposomes well and I am therefore not convinced of the main conclusion of the work.

Most strikingly in my eyes is the insufficiency of purity of the TAM. Fig. 2d shows that not only TamA and TamB but also multiple proteins were identified in the TAM proteoliposomes, including bands with similar molecular weights to BamA and BamD (see attached image). Since these bands were not identified in the TamA proteoliposome (Fig. S2C), it is possible that these proteins were contaminated with the addition of TamB. Previous reports have demonstrated that even the presence of BamA or BamD alone in liposomes has OMP assembly activity for those proteoliposomes (Hagan et al., 2010 Science). The worst-case scenario, therefore, is that the addition of TamB has led to the contamination of BamA or BamD or both, resulting in OMP assembly activity. However, the authors have not been able to provide sufficient data to dispense with this worst-case scenario. To dispel this possibility, the authors should show the absence of the insertase activity in assembly experiments with proteoliposomes prepared using a loss of function mutant of TAM.

As we note in the text (lines 197-198), an anti-BamA antiserum only recognizes a band in BAM/PLE proteoliposomes and not in TAM/PLE proteoliposomes (Fig. 2d, lanes 9-10). We have increased the exposure and reduced the figure contrast to make this point clearer.

Nevertheless, to address the reviewer's concern we performed three new experiments (new Supplementary Fig. 4). We measured the amount of BamA and BamD that is present in the TAM and TamA proteoliposomes (Supplementary Fig. 4a). When we load massive amounts of TAM/PLE or TamA/PLE onto a gel, we do in fact see a very small amount of BamA and BamD, but the BamA contaminant represents <0.1% of the total protein and the BamD contaminant represents <0.7% of the total protein. We show in Supplementary Fig. 4b that this tiny amount of BAM would not be sufficient to catalyze OMP assembly. Furthermore, we show in Supplementary Fig. 4c that TAM-dependent OmpA assembly, but not BAM-dependent assembly, is inhibited by PMSF (as we note, PMSF also did not appear to significantly affect BAM activity in the studies described in refs. 78 and 79). This result confirms that TAM, and not BAM contaminants, accounts for the assembly of OmpA in our assays. We have changed the text accordingly to cite our new results (lines 246-252). As an aside, we believe that many of the "contaminants" in our TAM preps are TamB breakdown products because they are larger than TamA, they increase if the TAM/PLE proteoliposomes are stored at -80°C for over three months, and they follow the complex through the reconstitution step (see Fig. 2b, lane 3).

Finally, we should note that we are not aware of any TAM loss-of-function mutants (which would have to contain point mutations rather than large deletions) that could be used to easily dispel the "worst-case scenario". A few double mutations that "lock" the TamA lateral gate closed have been recently described that appear to only modestly affect the function of TAM *in vivo* (ref. 43). We purified TAM containing these mutations in TamA and found that the mutations strongly

disrupted the interaction between TamA and TamB (only a small amount of TamB co-purified with TamA) and therefore could not be used for further experiments.

The authors propose in Fig. 7 that TamB interaction with the substrate facilitates TAM-mediated assembly. However, no data are presented in this paper that indicates a direct interaction between TamB and the substrate. Because previous studies have shown that TamB stimulates conformational changes in TamA (Shen et al., 2014 Nat Commun), the reasons why TamB enhances the insertase activity of TamA are not limited to the molecular mechanism proposed by the authors.

Although the cartoon in the original version of Fig. 7 effectively shows only one possible scenario, we believe that the reviewer raises a very good point. To address his/her concern, we have now modified the Discussion (lines 490-491 and 494-498) to indicate that because there is no clear evidence that TamB binds to OMPs, chaperones might target specific OMPs to TamA and then, perhaps by interacting with the TamA POTRA domains, cause a conformational change in TamB that leads to the opening of the "lateral gate" or the activation of TamA by another mechanism. We have now redrawn Fig. 7 to show both possible scenarios and modified the Figure legend accordingly. We have also removed the phospholipids from the cartoon because we believe that they may confuse readers.

Since Nature Communication is a general journal targeting a broad readership, I believe that confirmation under physiological conditions as well as validation in *in vitro* assays is essential. If the authors claim that TAM has the same insertase function as BAM, the authors should provide results of experiments showing that the effects of loss of function mutants of the BAM can be suppressed by overexpression of the TAM.

In principle, we agree with the reviewer that it would be fantastic if we could confirm our results under physiological conditions. We have found, however, that an experiment that would confirm our results *in vivo* is not feasible. The first problem is that we are unaware of a "loss-of-function" mutant (e.g., a conditional mutant) that we know would allow us to inactivate BAM activity rapidly. We previously showed that darobactin completely blocks BAM activity *in vivo* within one minute of addition (ref. 37), but we cannot use this compound to perform the type of experiment that the reviewer suggests because darobactin also inhibits TAM (Fig. 6e). In addition, BAM catalyzes the assembly of TAM (ref. 94), and inhibiting BAM activity would likely disrupt TAM biogenesis. Nevertheless, in an attempt to address the reviewer's concern, we decided to use a *bamA101* strain that produces 5-10 fold less BamA than wild-type strains as a proxy for a loss-of-function mutation. Indeed it has been shown that *bamA101* strains grow relatively well, but are defective in OMP assembly (Aoki SK et al. (2008) Mol. Microbiol. 70: 323-340). We introduced plasmids into a *bamA101* strain in which *tamAB*, *tamA* or *bamA* is under the control of an inducible promoter, but discovered that even in the absence of inducer the *tamAB* plasmid (but not the *tamA* or *bamA* plasmid) is very toxic. Although we tried to determine if overexpression of *tamAB* would rescue the assembly of OmpA, after we added the inducer the cells stopped growing and TamA could not be detected at later time points on a Western blot. In contrast, the cells that contained the *bamA* plasmid grew well and we could detect high levels of BamA. Of course, *tamAB* overexpression could be toxic for many different reasons.

On another level, we would like to note that the main goal of our study was to develop an assay that would enable us to determine if purified TAM can catalyze OMP assembly. We wanted to see if we could obtain *direct* biochemical evidence that complements previously published

genetic evidence that *indirectly* implicates TAM in OMP assembly. Given that the function of TAM has recently become less clear, we believe that our results represent a significant advance in the field that should be of broad interest. We not only show that TAM is involved in OMP biogenesis, but we also show that it can independently catalyze OMP assembly nearly as well as BAM. The latter observation is particularly important because it helps to rule out plausible models in which TAM functions only as a BAM accessory factor. Finally, the finding that TamA and BamA use similar mechanisms to catalyze OMP assembly is also important, because that conclusion could not have been reached based on previously results alone.

Reviewer #2 (Remarks to the Author):

Gram-negative bacteria have an outer membrane outside of their peptidoglycan layer that provides an essential layer of protection against environmental toxins and insults. However, this membrane is far removed from the bacterial cell's protein and lipid biosynthetic machinery and therefore the biogenesis of the outer membrane is a challenging cellular process. The beta-barrel assembly machinery is responsible for the assembly of the vast majority of outer membrane proteins (OMPs) and is composed from a core BAM complex and a translocation and assembly module (TAM) that together ensure that all beta-barrel proteins are assembled with optimal efficiencies.

Over the past 10 years, a great deal of work has been invested into understanding these assembly machines, with the BAM complex having the most attention. In the case of the TAM it is clear from kinetic studies conducted *in vivo* that the TAM adds to the speed and efficiency of a number of OMPs. The TAM has been purified and reconstituted into membranes to demonstrate that the TAM can mediate the insertion of an OMP into the membrane layer (for example, measured by neutron reflectometry), but it has never been demonstrated whether the purified TAM can mediate the folding of OMPs into their final, native form. In this manuscript by Wang et al, that proof is finally provided and the assay system created for that purpose is used to monitor the folding of 4 OMPs, each with distinct extracellular domains in addition to their membrane-integral beta-barrel domain. This is a powerful assay and it is used to great effect. The quality of the data is excellent and I have no suggestions for additional wet-lab experimental work. The major short-coming of the paper is considerable inaccuracy when it comes to crediting past work published by other research teams and an inaccurate over-emphasis on both what was missing in our knowledge and in seeking to propose controversies and complications that do not exist in the literature. The specific suggestions for addressing these short-comings, some minor and some major, are detailed below.

We thank the reviewer for his/her positive comments.

1. Title: please replace the name "Tam complex", which is not used in the literature, with the accepted name "TAM" or the long form Translocation Assembly Module. This also provides for internal consistency as TAM is used elsewhere in the paper (other than the legend of Figure 1).

We changed the name from "Tam complex" to "translocation assembly module (TAM)" in the title. To meet the journal's 15 word limit we also had to remove "purified *E. coli*" from the title.

2. Abstract (lines 29-30): Please correct the statement "Although its function is controversial, TAM has been proposed to play a critical role in the assembly of a small subset of *E. coli* OMPs

based primarily on experiments conducted in vivo using tamA and/or tamB deletion or mutant strains". This statement is not factual. There is no controversy about the function of the TAM with published evidence from many papers showing that it functions in the assembly of several types of OMPs (autotransporters, inverse autotransporters, fimbrial ushers and efflux pumps; not all of this evidence is cited in the current manuscript). This evidence has included experimentation both in vivo - using tamA and/or tamB deletion or mutant strains - as well as in vitro reconstitution experiments using quartz crystal microbalance measurements and/or analysis by neutron reflectometry.

To address this concern we have revised the Abstract. Most notably, we have removed the statement that the function of TAM is controversial. To maintain accuracy, we now state that there is evidence that TAM catalyzes OMP assembly as well as evidence that it helps to maintain phospholipid homeostasis. With respect to quartz crystal microbalance measurements and neutron reflectometry that show that the binding of antigen 43 to TAM leads to conformational changes in TamA, we have noted in the Abstract that the results of biophysical experiments have also provided evidence that TAM is an insertase (lines 30-31) and cite the appropriate papers (refs. 44 and 46) in the Introduction and the Discussion (line 419). To provide a more comprehensive list of proteins that require TAM for efficient biogenesis *in vivo*, we now mention UshC and TolC in the Introduction (line 107) and have added new references (44, 46, 47, 52, and 53).

3. Introduction (lines 120-121) Please correct the statement "To complicate matters, however, recent evidence strongly suggests that TamB is a member of the AsmA-like family of proteins and works with TamA to promote phospholipid transport to and from the OM of *E. coli*[59,60]."

That the TAM might also play a role in phospholipid transport in the absence of the major phospholipid transporter (YhdP) is neither a complication nor controversial. That TamB is a member of the AsmA-superfamily of proteins is not a new observation, and both References [59 and 60] cite (PMID: 25994932) as the source of these observations and for the characterization of the AsmA-related groups of proteins including YhdP, TamB, and YdbH.

We changed this sentence to "It is very noteworthy, however, that recent evidence strongly suggests that TamB, which is a member of the AsmA-like family of proteins⁴⁰, works together with TamA to promote phospholipid transport to and from *the E. coli* OM^{66,67}" (lines 119-122). To correct our error, we now cite PMID: 25994932 (ref. 40) as the source for the discovery that TamB is a member of the AsmA-like protein family.

4. (Lines 124-126, as per point 1): In order to correctly acknowledge the previous works on TamB, it would be important to refine the text "To overcome the potential problem of indirect effects of TAM mutations in studies performed in vivo and to test the ability of TAM to catalyze OMP assembly directly, we developed a novel method to purify and reconstitute TAM into lipid vesicles in vitro".

A more accurate statement of the contribution would be to recognize that the current study is the first to show that the TAM is necessary and sufficient to fold the beta-barrel domains of four OMPs in a purified system, without the assistance of the BAM complex.

To address this concern we changed the first sentence of the last paragraph of the Introduction to "To test the ability of TAM to catalyze OMP assembly independently from BAM..." (line 124). We also removed "To overcome the potential problem of indirect effects of TAM mutations in

studies performed *in vivo* and to test the ability of TAM to catalyze OMP assembly directly...". Later in the paragraph we note that our results show that TAM is sufficient to catalyze OMP assembly, but we think that "necessary" might be confusing because we show that the OMPs we tested can also be assembled by BAM.

5. Figure 1. (lines 143-147): "In superimposing the published structures of the *E. coli* BamA and TamA β barrels^{44,61}, we noticed a remarkable degree of overlap in the two β barrel backbones that has been previously reported⁶² (Fig. 1A). Although most of the loops in the TamA β barrel are shorter than their cognate loops in the BamA β barrel, the fundamental similarity in overall structure suggested a similarity of function."

As noted in this text, these are published observations by other authors. Why is it being published again in this paper? This text (and Figure 1B) should be removed. Figure 1B is an excellent figure to start the paper.

We believe that the superimposition provides important information that will help a general audience understand why we think that TamA and BamA have similar functions. Many papers show previously published structures to aid readers who may not have seen the structures before. Nevertheless, to properly credit published studies, we have rewritten the first paragraph of the Results. We now state that "A remarkable degree of overlap in the *E. coli* BamA and TamA β barrels was previously observed in a study in which published crystal structures were superimposed⁴⁷ (Fig. 1a)" (lines 142-143). To further document the similarity of the two β barrels, we have also added an RMSD analysis to the Supplement (Supplementary Table 1; lines 145-146).

6. Figure 1 legend: please replace the word "Tam complex or TAM" with the accepted name "TAM". This provides for internal consistency as TAM is used elsewhere in the paper.

As suggested, we replaced "Tam complex" with "TAM" (Line 1004).

7. Figure 2. The data is excellent and convincing, but the protein bands are barely visible and the background is too white (i.e. the contrast on the images is too high, potentially obscuring other species in the gels). The figures should be adjusted to rectify this issue.

As suggested, we reduced the contrast of all of the Western blots in Fig. 2.

8. Figure 5, Figure 6. The primary data is excellent and convincing. I was not able to find any reference to how the "% cleaved" and "% folding" figures were calculated, and no errors are noted – is there no margin of error in these measurements (for example, multiple quantifications of the same data and/or normalization corrections on the multiple experiments)?

To improve the quality of the data, we have now repeated each of the experiments shown in Figs. 5 and 6 several times and provide statistics in Supplementary Figs. 7 and 9 (new Figures) to demonstrate the reproducibility of the results. We also explain how we calculated "% cleaved" and "% folded" in the legend to Fig. 4 and Supplementary Fig. 4 (another new Figure that shows the results of experiments suggested by reviewer 1).

9. Figure 7. Why is TamB represented as a half-pipe emphasizing lipid transport here, and yet represented as a superhelical twist in Figure 1? The scale for these two figures is important, can they be adjusted so that the scale is (approximately) similar in the structural perspective (Figure 1) and the cartoon representation (Figure 7).

We used a modified version of the three-dimensional structure of *E. coli* TamB predicted by AlphaFold2 and MD simulations (ref. 95) to generate Fig. 1b, but in Fig. 7 TamB is shown only as part of a cartoon. Nevertheless, to address the reviewer's concern we have modified both Fig. 1b and Fig. 7 to increase the consistency of our depictions of TamB. Because the predicted distance between the two ends of the TamB β -taco (G27 and D1135) is only 125 Å, which is much shorter than the estimated width of the periplasm (~250 Å) the authors of ref. 95 proposed an optimized structure of TamB that is less twisted and would therefore traverse the periplasm. We cite ref. 95 in the legend of Fig. 1b to provide the source of the predicted structure. We have drawn TamB in Fig. 7 as a twisted helix to match the predicted structure, but it should be kept in mind that the purpose of Fig. 7 is to show a conceptual model in cartoon form rather than to focus on accurate structural information.

10. Figure 7. It should be made clear in the Legend or the main text that the binding of substrate protein into the beta-taco groove, as well as the presence of lipids in the beta-taco groove, are purely speculation. Neither of these events has been experimentally documented.

To address the reviewer's concern (and a similar concern raised by reviewer 1) we have now removed the phospholipids from the cartoon. We also mention in the Discussion (lines 494-498) that because there is no evidence that TamB binds to OMPs, it is possible that chaperones target OMPs to TamA and, perhaps by binding to the POTRA domains, trigger a conformational change in TamB that initiates the assembly process. We have now redrawn Fig. 7 to try to show two alternative scenarios that both seem reasonable, one in which OMPs bind to the β -taco groove of TamB, which then targets them to TamA, and another in which SurA or other conventional chaperones target OMPs directly to TamA and thereby promote conformational changes in TamB that in turn activate TAM. To address the reviewer's concern further, we also mention in the Discussion that there is no direct evidence that phospholipids bind to TamB (line 516). Finally, we modified the legend to Fig. 7 and now state that "Possibly due to partial folding and the exposure of a hydrophobic surface, a subset of OMPs *might* subsequently bind to TamB, which uses the hydrophobic interior of its β -taco structure to escort them to TamA..." (lines 1100-1102) and added that the proposed anterograde transport of phospholipids occurs "*presumably* through the interior of the β -taco" (line 1109).

11. Supplementary Figure S1. In the absence of heat, the purified TAM migrates at a size somewhere between 480kDa and 720kDa. What is the basis for this size? This should be noted in the main text of the paper.

The main reason we performed the blue native PAGE analysis was to help show that our purified TAM has the same properties as TAM purified by other investigators. As we note on line 180, the molecular weights that we observed are similar to those observed in previous studies (refs. 38 and 44). We believe that most readers will know that proteins and protein complexes often do not run at their predicted molecular weight on blue native PAGE. To address the reviewer's concern, we also added more details about the native PAGE experiments and indicate that the molecular weights were determined using the Thermo Fisher NativeMark Unstained Protein Standard in the legend to Supplementary Fig. 1.

12. Supplementary Figure S1. In the presence of heat, TamA migrates at a size somewhere between 148kDa and 242kDa, despite being a protein of 50-60kDa in size. What is the basis for this size? This should be noted in the main text of the paper.

Please see our response to comment 11. To address the reviewer's concern further, we note in the legend to Supplementary Fig. 1 that if we run TamA against the NativeMark standard after heating the protein migrates at ~150 kDa, but if we run it against conventional SDS-PAGE molecular weight markers (LICOR Chameleon Duo pre-stained protein markers) it migrates at ~65 kD (see below). Although we are providing this information to the reviewers, we thought that it might confuse readers if we showed TamA run against two different types of molecular weight markers on the same gel in our manuscript. Nevertheless, as we state in the legend to Supplementary Fig. 1, we have included the gel in the Source Data file for anyone who is interested in seeing it.

13. Supplementary Figure S3. I leave this to the discretion of the authors: could you consider making Fig S3A part of main Figure 3? I would have found this summary information helpful when reading the main text.

We would prefer to leave the structures in the Supplement because the data in Fig. 3 is a critical part of the story that we believe should stand alone. If we add the structures to Fig. 3 we will have to reduce the size of the Western blots. Furthermore, we are concerned that by denoting both known and potential cleavage sites in the OMPs (as we do in Supplementary Fig. 3a) we might confuse some readers.

14. Supplementary Figure S6/Table S1. The reasoning for the selection of the species for which the sequence analysis is done is hard to reconcile. Table S1 notes that some of the species do or don't have all three Omp85 types in their genomes. Why not choose species which do have all three types for the analysis? Also, why is the analysis done with so few protein sequences? The motifs that are generated are highly biased as a result. In the current study, the size of the conserved letters here would suggest almost strict conservation across species. However, this is not what is found in previously published work where a comprehensive analysis of all available sequences was used. In the previous unbiased analyses, relatively few residues are highly conserved, and none are absolutely conserved.

To address the reviewer's concern, we have modified our sequence analysis. We have now chosen one member of each of the 39 Proteobacterial families in which TamA has been annotated in the Uniprot database for our analysis. The reason we chose only one member of each family is to try to minimize bias; after all, some families (e.g., Enterobacteriales) are highly overrepresented. We now state on lines 343-347 that "...in members of the 39 Proteobacterial families in which TamA has been annotated, TamA and BamA share a conserved sequence motif...", show a revised logo plot in Supplementary Fig. 8, and list all of the species that we used in our analysis in Supplementary Table 2a. We also selected one TpsB homolog from each Proteobacterial family for our analysis. In cases where there are no annotated TpsB homologs in the species that we used for the BamA/TamA analysis, we selected a related species from the same family in which a TpsB homolog is annotated (Supplementary Table 2b).

Although our analysis of the conservation of sequence motifs has inherent limitations, the similarity of the *E. coli* BamA and TamA sequences that we show in Fig. 6a is much more important because we only compare the activity of *E. coli* BAM and TAM in our experiments. We should also note that unlike previous investigators, we did not perform a comprehensive analysis of all available sequences, especially since TamA is produced primarily in Proteobacteria and the Bacteroidetes, and it is certainly possible that homologs produced in the distantly related Bacteroidetes (which can be only partially aligned to *E. coli* TamA by BLAST) play a very different role in OM biogenesis and membrane homeostasis.

15. Discussion. In my opinion, the Discussion would be greatly improved by contrasting what is shown herein that focusses on the correct folding of OMPs mediated by the TAM (the novelty of this work) with the previous demonstration that the reconstituted TAM mediated the insertion of a model OMP (i.e. the equivalent of EspP) (PMID: 25341963 and PMID: 26243377).

To address the reviewer's concern we now state on line 419 that Ag43 has been previously shown to initiate dynamic movements of TAM in a reconstituted system (refs. 44 and 46). Based on our reading of ref. 44 (PMID: 25341963) and ref. 46 (PMID: 26243377), neither paper shows that TAM mediates the membrane insertion of a model OMP.

Reviewer #3 (Remarks to the Author):

Review Response for The purified *E. coli* TAM complex catalyzes the assembly of bacterial outer membrane proteins in vitro

Wang et al. present the first example of OMP folding in vitro by the TAM complex. After expression and purification of the TAM complex (consisting of TamA and TamB), they demonstrate that TAM can fold four OMPs of varying sizes into proteoliposomes. They suggest that this is via a mechanism that is similar to the BAM assisted folding mechanism because addition of darobactin or Skp and mutation of the β -signal all reduce folding in a manner similar to their action on BAM.

This is a timely paper as the evidence for TAM function is unclear and based on indirect genetic evidence. This manuscript reflects that TAM can fold OMPs, but it remains unanswered as to whether this is the primary function of TAM and if TAM is involved in lipid homeostasis.

We thank the reviewer for his/her positive comments.

Some key points to address include:

- McDonnell et al. (bioRxiv 2023) have used AlphaFold2 multimer to predict the complex between TamA and TamB. It suggests that TamB interacts with TamA at the 'lateral gate' by forming a beta augmented hybrid barrel. How then can TamA fold OMPs without first dissociating from TamB? Could the authors please comment on their model for TAM mediated folding in the light of this model structure? Do the authors have any evidence that TamA and TamB are still interacting in the beta signal variants or in the samples where darobactin has been added?

While the preprint that the reviewer mentions is certainly very interesting, it is still not clear how TamA and TamB interact. Several previously published experimental studies have used a combination of the TamA N-terminal POTRA domain and a TamB C-terminal peptide, TamA POTRA-domain deletion mutants, and a BamA β barrel-TamA POTRA domain chimera to obtain evidence that TamB C-terminus interacts with the first POTRA domain of TamA *in vitro* and *in vivo* via a putative β -augmentation mechanism (refs. 38, 44, 46, and 47). We now refer to these studies in the Introduction (lines 101-103). The AlphaFold2 prediction described by McDonnell et al. which suggests that TamB interacts with TamA at the "lateral gate", however, looks very convincing. It is certainly possible that both the experimental evidence and the structure prediction are correct. Perhaps TamB binds to different segments of TamA at different stages of OMP assembly. Indeed in the inactive state TamB might bind to the first POTRA domain of TamA, but then in the active state adopt a less twisted conformation and interact with the TamA lateral gate to serve as a placeholder for an incoming OMP. Alternatively, when TamB performs different functions (i.e., phospholipid transport versus OMP biogenesis) it might bind to different regions of TamA. We now cite the preprint, but note that "It is unclear...whether TamB...bind[s] to different TamA domains to mediate two distinct biological processes or to facilitate a transition between two different functional stages of a single process." (lines 523-527).

In an effort to determine whether TamB dissociates from TamA during OMP assembly, we performed an experiment in which we first bound DDM solubilized and purified TAM (that contains a His tag on TamA) to nickel beads (TAM/PLE proteoliposomes do not bind to the nickel resin). We then resuspended the beads in a small volume and added urea denatured OmpA. We did not observe any release of TamB from the beads under our experimental conditions. We believe that this negative result is difficult to interpret and should not be included in our manuscript.

- The examination of the addition of specific chaperones with TAM and BAM is interesting, but is it not surprising that the addition of SurA to the BAM sample does not increase the folding of the substrate OMP given that previous papers have shown that SurA increases folding of OmpA variants by between 50-80% (Devlin et al 2023, Schiffrin et al 2022). The experiment on chaperones and TamAB was only performed with EspP. Could this be done with OmpA too?

As suggested by the reviewer, we have performed experiments that involve the addition of specific chaperones to OmpA assembly reactions (see Fig. 5 and Supplementary Fig. 7, which shows a statistical analysis of three independent experiments) and modified the text accordingly to add the new data (see lines 325, 327-330).

- In Figure 1b when TAM is assembled in a proteoliposome, the TamB transmembrane helix does not appear to be assembled into a liposome, would this not be the case?

In the interest of clarity we now state in the legend to Fig. 1 that it is unclear whether the TamB N-terminus is integrated into a liposome (line 1008). It is also unclear whether integration of the N-terminal α -helix into a liposome would be important for TamB function in our *in vitro* assays. Because there are no lysines or arginines in the α -helix, the only fragment of the protein that would be protected from trypsin digestion if TamB were integrated into a liposome would be the N-terminal five residues that precede the α -helix, which would be impossible to resolve from other tryptic fragments by SDS-PAGE. Fig. 1b is simply a cartoon that is based on our trypsin digestions (Fig. 2c) and previously published evidence that TamB interacts with the POTRA domains of TamA (refs. 38, 44, 46, and 47).

- The TamA structural 'predictions' are strange – simply aligning a beta strand next to another will create H-bonds between backbone atoms and does not indicate that it binds. Terming this in the methods as Model building and refinement suggests that there is experimental data to refine into which is not correct. Given the information available, structural alignments are only possible and the ability to say that the binding of darobactin or substrate does not cause any major clashes. In figure 1, the alignments between TamA and BamA would benefit from RMSD values and would move the text away from terms such as the barrels are similar (line 433)

To address this concern, we changed the term “Model building and refinement” to “Structural alignments and predictions” in the Methods section (Line 689). We also modified both the main text to indicate that seven backbone H-bonds can form “without causing any steric clashes” (Line 352) and the legend to Fig. 6b to point out that the binding of TamA (β 1) to EspP (β 12) was predicted based on the indicated BamA(β 1)-EspP(β 12) structural model (Line 1071).

As also suggested by reviewer 2 (comment 5), we calculated six different RMSD values for TamA and BamA β barrels and listed them in Supplementary Table 1. As we now mention on line 145, the backbones of the TamA and BamA β -strands have an RMSD of 2.078 Å.

Reviewer #1 (Remarks to the Author):

In my assessment of the manuscript titled “The translocation assembly module (TAM) catalyzes the assembly of bacterial outer membrane proteins *in vitro*” by Wang et al., I raised three concerns. The authors effectively addressed my first concern through explanations and additional experiments. However, my second and third concerns remain unresolved as they lack confirmation *in vivo*.

Regarding the second point, which pertains to Figure 7, the authors graciously accepted and rectified my observation. However, my initial suggestion was predicated on the assumption that an *in vivo* analysis, alluded to in the third point, would be conducted. Unless this analysis directly addresses the third concern, it necessitates further revision. Notably, the manuscript lacks conclusive experimental evidence that the TAM complex independently assembles outer membrane proteins (OMPs) *in vivo*. Consequently, the schematic in Figure 7 should depict proteo-liposomes, akin to Figure 1, rather than an *in vivo* context. Descriptions such as “inner membrane” and “sec translocon” are therefore inappropriate.

The third concern pertains to generalization without *in vivo* analysis. To address this, I propose utilizing the BamA depletion strain, a mutant known to rapidly inactivate BAM activity (Lehr U., et al., 2010, Mol Microbiol.). By incubating this strain in the presence of glucose for approximately 3 hours, the OMP levels decrease due to reduced BamA. In just 5 hours, BamA is entirely depleted. This strain could serve as a valuable tool to assess whether TAMs indeed assemble OMPs *in vivo*. If overexpression of TAMs became toxic as authors’ mentioned, alternative strategies include expressing them with a constitutive promoter to mitigate toxicity or expressing either TamA or TamB. The authors’ cited challenges can potentially be circumvented using these approaches. If technical limitations persist, it is essential to acknowledge them as a limitation in the study.

Reviewer #2 (Remarks to the Author):

The authors have addressed the issues that I raised in the first round of review.

Reviewer #3 (Remarks to the Author):

outer membrane proteins *in vitro*

Wang et al, have tried to address some of the reviewers concerns regarding this paper. However, I do not believe all the concerns raised during revision have been addressed.

I am not convinced that the proposed mechanism is supported by the evidence and, worse, could well be incorrect.

1. The authors provide no direct evidence as to how TamA and TamB interact. They cite papers that use chimeras or massively truncated proteins to suggest that TamB interacts with POTRA-1. The evidence in the current manuscript for this interaction is that no band is visible in trypsin digest for TamB, but the gel is cropped out at 15kDa. An interaction of TamA and TamB as proposed by the AlphaFold model would probably be indicated by a band for TamB at a size smaller than 15kDa (if it could be observed on a gel and was a single digested product). Furthermore, in response to reviewer 1, the authors state that mutations that lock TamA in a 'closed' state 'strongly disrupted the interaction between TamA and TamB (only a small amount of TamB co-purified with TamA)'. If as the authors suggest TamA interacts with TamB at POTRA-1 why would locking the barrel shut stop co-purification? If anything closing the barrel is necessary to their proposed mechanism.

The Alphafold model strongly predicts a TamA-TamB interaction that is at complete odds with the model presented. Give that this is the key to the submitted manuscript I do not think the manuscript should progress without firm evidence to show the Alphafold model is incorrect. This manuscript presents a clear model for TamAB in Figure 7 that could be entirely wrong.

2. The authors show that the addition of chaperones to TAM proteoliposomes or to TamA has no effect on the amount of folded OMP. So why does the model in Fig 7. show that chaperones can deliver to TamB or TamA. The authors present no evidence that this is the case.

3. No discussion is made about what happens in the absence of TamA. Why was no control in the absence of TamB completed? Can TamB be purified in the absence of TamA?

REPLY TO REVIEWERS' COMMENTS (all major changes in the text are highlighted)

Reviewer #1 (Remarks to the Author):

In my assessment of the manuscript titled "The translocation assembly module (TAM) catalyzes the assembly of bacterial outer membrane proteins *in vitro*" by Wang et al., I raised three concerns. The authors effectively addressed my first concern through explanations and additional experiments. However, my second and third concerns remain unresolved as they lack confirmation *in vivo*.

Regarding the second point, which pertains to Figure 7, the authors graciously accepted and rectified my observation. However, my initial suggestion was predicated on the assumption that an *in vivo* analysis, alluded to in the third point, would be conducted. Unless this analysis directly addresses the third concern, it necessitates further revision. Notably, the manuscript lacks conclusive experimental evidence that the TAM complex independently assembles outer membrane proteins (OMPs) *in vivo*. Consequently, the schematic in Figure 7 should depict proteo-liposomes, akin to Figure 1, rather than an *in vivo* context. Descriptions such as "inner membrane" and "sec translocon" are therefore inappropriate.

We would like to emphasize that Fig. 7 only shows a model based on our *in vitro* experiments and previous *in vivo*, *in vitro* and *in silico* studies. We believe that especially for readers who are not experts in the field it is preferable to show TAM in an "*in vivo* context" to illustrate the big picture of OMP assembly. Furthermore, we know that molecular chaperones are required for OMP assembly *in vivo* and that at least one chaperone (PpiD) interacts with the Sec complex, so we do not think it is unreasonable to show a chaperone bound to the Sec complex that, at least in one possible scenario, directs OMPs to TAM. Nevertheless, to address the reviewer's concern, on line 416 we now state that we developed an *in vitro* assay that we used to "analyze" TAM function instead of "to reproduce" TAM function, and on line 493 we now state that our model is based on both previous studies and our "*in vitro* results".

The third concern pertains to generalization without *in vivo* analysis. To address this, I propose utilizing the BamA depletion strain, a mutant known to rapidly inactivate BAM activity (Lehr U., et al., 2010, Mol Microbiol.). By incubating this strain in the presence of glucose for approximately 3 hours, the OMP levels decrease due to reduced BamA. In just 5 hours, BamA is entirely depleted. This strain could serve as a valuable tool to assess whether TAMs indeed assemble OMPs *in vivo*. If overexpression of TAMs became toxic as authors' mentioned, alternative strategies include expressing them with a constitutive promoter to mitigate toxicity or expressing either TamA or TamB. The authors' cited challenges can potentially be circumvented using these approaches. If technical limitations persist, it is essential to acknowledge them as a limitation in the study.

As we noted previously, several studies on the function of TAM *in vivo* have already been conducted by other groups and rely on genetic manipulations, all of which have inherent shortcomings. Our goal was to examine the function of TAM in a biochemical assay that used purified components. Biochemical assays have inherent shortcomings as well, but to confirm that a cellular factor has a specific function both the genetic and biochemical results should agree. The Tommassen and Silhavy labs showed in papers published in *Science* and *Nature* that the depletion of BamA led to a defect in OMP assembly, but it was not until Dan Kahne's lab showed in a separate *Science* paper several years later that purified BAM could catalyze OMP assembly *in vitro* that it became clear that BAM plays a direct role in the assembly

reaction. In essence we are extending the results of previous *in vivo* studies by showing that purified TAM catalyzes OMP assembly *in vitro*.

We agree with the reviewer that it would be fantastic if we could show that TAM can substitute for BAM *in vivo*. So far, no other group has been able to draw that conclusion. For this reason, we performed the experiment that he/she suggested (see below), although we used a different BamA depletion strain (JM166, which was generated by the Silhavy lab). To effectively deplete BamA we found that it was necessary to grow the cultures to saturation in the absence of arabinose, redilute them, and grow them to mid-log phase (this method was most recently used by Wu R. *et al.*, *Nat. Comm.* 2021, 12: 7131 PMID: 34880256). We discovered that the expression of TAM or TamA during BamA depletion was highly toxic and that the expression of TamB was only slightly toxic (see ODs in part A; the expression levels of BamA, TamA and Ffh—a loading control—are shown on the Western blot in part B). Interestingly, we also found that the expression of TAM partially restored the assembly of OmpA, Ag43 and FadL (part C, lane 5) and even BamA (part B, lane 3—but that the expression of TamB alone restored assembly more effectively (see part B, lane 5, top blot and part C, lane 9). Especially in light of the toxicity of TAM expression, this result is ambiguous and can be interpreted in different ways. Is TAM truly able to replace BAM, or do both TAM and TamB promote the assembly of OmpA and BamA by an unknown mechanism? There is no way to distinguish between these two possibilities.

Because technical problems have persisted, to address the reviewer's concern we now state on lines 485-489 that a limitation of our study is that we could not determine if TAM can replace BAM *in vivo*.

Reviewer #3 (Remarks to the Author):

Wang et al, have tried to address some of the reviewers concerns regarding this paper. However, I do not believe all the concerns raised during revision have been addressed.

I am not convinced that the proposed mechanism is supported by the evidence and, worse, could well be incorrect.

1. The authors provide no direct evidence as to how TamA and TamB interact. They cite papers that use chimeras or massively truncated proteins to suggest that TamB interacts with POTRA-1. The evidence in the current manuscript for this interaction is that no band is visible in trypsin digest for TamB, but the gel is cropped out at 15kDa. An interaction of TamA and TamB as proposed by the AlphaFold model would probably be indicated by a band for TamB at a size smaller than 15kDa (if it could be observed on a gel and was a single digested product). Furthermore, in response to reviewer 1, the authors state that mutations that lock TamA in a 'closed' state 'strongly disrupted the interaction between TamA and TamB (only a small amount of TamB co-purified with TamA)'. If as the authors suggest TamA interacts with TamB at POTRA-1 why would locking the barrel shut stop co-purification? If anything closing the barrel is necessary to their proposed mechanism.

The Alphafold model strongly predicts a TamA-TamB interaction that is at complete odds with the model presented. Give that this is the key to the submitted manuscript I do not think the manuscript should progress without firm evidence to show the Alphafold model is incorrect. This manuscript presents a clear model for TamAB in Figure 7 that could be entirely wrong.

We are very grateful that the reviewer raised this issue again because after further thought, we agree that even though the new AlphaFold2 prediction of the interaction of TamA and TamB has not yet been published, we should include it in our model. To address the reviewer's concern we now describe the *in silico* evidence that the C-terminus of TamB folds into a partial β barrel-like structure that binds to TamA β 1 in the Introduction and cite the pre-print by McConnell et al. (lines 103-107). Second, to provide experimental evidence that supports the AlphaFold2 prediction, we have now included the data that shows that only a small amount of TamB co-purifies with a laterally "locked" form of TamA (see new Fig. S3 and lines 197-202). Third, we have changed our model by proposing that the C-terminus of TamB acts as a placeholder for incoming OMPs (Discussion, lines 494-498) and by significantly modifying Fig. 7 and the legend to Fig. 7 (lines 1112-1119).

As suggested by the reviewer, we have repeated the trypsin digest and loaded 20x the amount of protein shown in Fig. 2c on a gel that is better suited to resolve small products (see below). Although we do not see a ~8 kD band that might correspond to the C-terminus of TamB, it is possible that by digesting the POTRA domains the trypsin causes the release of the TamB C-terminus from the TamA lateral gate. We also do not see a ~2.5 kD band that might be derived from the TamB N-terminus. Based on this result we have not changed the statement in the legend to Fig.1b in which we indicate that it is unclear if the N-terminus is reconstituted into a separate liposome, but we have included the results of the new trypsin digest shown below in the Source Data file.

2. The authors show that the addition of chaperones to TAM proteoliposomes or to TamA has no effect on the amount of folded OMP. So why does the model in Fig 7. show that chaperones can deliver to TamB or TamA. The authors present no evidence that this is the case.

We did not address this concern because it was not raised previously. Nevertheless, we should point out that when we used the same procedure to purify TAM and BAM, BAM did not require chaperones either (Fig. 5), and yet it is widely believed that chaperones are required for BAM-mediated OMP assembly *in vivo*. Furthermore, it was shown in ref. 83, Fig. 4, bottom gel that BAM catalyzes OMP assembly in the absence of chaperones even when the complex is purified using a “standard” protocol, although the efficiency of assembly is clearly reduced. We believe the problem is that our *in vitro* assay—like virtually all other cell-free assays—does not perfectly replicate the events that occur in living cells. Based on the available evidence, it seems likely that at least one chaperone interacts with OMPs that are assembled by TAM *in vivo*. Although for the sake of simplicity we do not discuss PpiD in our manuscript, this chaperone binds directly to the Sec complex and has been shown to interact with OmpA as the protein emerges from the translocation machinery (see, for example, Antonoeae, R. et al. (2008) *Biochemistry* 47: 5649-56.)

We should also mention that reviewer 2 correctly noted that there is no direct evidence that OMPs bind to the TamB β taco groove. To address his/her concern, we revised our model to include the possibility that OMPs are targeted directly to TamA and that TamB functions primarily as a regulator of TamA conformation (see lines 495-498). If this hypothesis is correct, it would be extremely likely that OMPs would be targeted to TamA by a chaperone.

3. No discussion is made about what happens in the absence of TamA. Why was no control in the absence of TamB completed? Can TamB be purified in the absence of TamA?

Although the reviewer did not raise this concern previously, we did try to purify His-tagged TamB alone and six different His-tagged TamB fragments and found that either most of the protein in the Ni-NTA column eluates were contaminants or breakdown products or that we had to solubilize membranes using LDAO (the detergent used to purify a small C-terminal fragment of TamB in ref. 65), which we found diminished TAM activity *in vitro*. To address the reviewer's

concern, we now state that we were unable to purify TamB under conditions that would be compatible with our assays (lines 196-197). Of course we would not have been able to reconstitute TamB into lipid vesicles that are outer membrane surrogates, so it is unclear if we would have obtained interpretable results even if we could have purified TamB.

Reviewer #3 (Remarks to the Author):

The authors have made efforts to address the concerns raised by both referees. The work still leaves many questions unanswered as to how TAM actually folds OMPs, how TAM and BAM cooperate together, and the role of TAM in transporting lipid versus its role in folding OMPs in vivo. But the work is the first time TAM folding assays with OMPs have been reported. I would still suggest that the final figure is altered to including a lot of question marks, as it could be misunderstood by readers that do not look at the paper in depth to separate what the authors unequivocally show, and what they speculate.

REPLY TO REVIEWER'S COMMENT

Reviewer #3 (Remarks to the Author):

The authors have made efforts to address the concerns raised by both referees. The work still leaves many questions unanswered as to how TAM actually folds OMPs, how TAM and BAM cooperate together, and the role of TAM in transporting lipid versus its role in folding OMPs in vivo. But the work is the first time TAM folding assays with OMPs have been reported. I would still suggest that the final figure is altered to including a lot of question marks, as it could be misunderstood by readers that do not look at the paper in depth to separate what the authors unequivocally show, and what they speculate.

We agree with the reviewer that our work leaves many questions unanswered, and we have devoted a significant amount of space in the Discussion to reviewing the most salient issues. We believe, however, that adding a lot of question marks to the model shown in Fig. 7 would make the Figure very busy and very confusing. The title of the Figure ("Model of OMP assembly by TAM") should immediately notify readers that the cartoon contains a good deal of speculation and, like any model, should not be taken as proven fact. To address the reviewer's concern, however, we have slightly modified the legend to Fig. 7 to clarify that the events that occur after the binding of chaperones to nascent OMPs (line 1115) represent our best guess based on all of the available evidence. Of particular note, we present two different scenarios for the initiation of the assembly cycle using words such as "might" and "could" (lines 1116-1118).